# ProteinNPT: Improving Protein Property Prediction and Design with Non-Parametric Transformers

**Pascal Notin**[†][*]
Computer Science,
University of Oxford

**Ruben Weitzman**[†]
Computer Science,
University of Oxford

**Debora S. Marks**
Harvard Medical School
Broad Institute

**Yarin Gal**
Computer Science,
University of Oxford

## Abstract

Protein design holds immense potential for optimizing naturally occurring proteins, with broad applications in drug discovery, material design, and sustainability. However, computational methods for protein engineering are confronted with significant challenges, such as an expansive design space, sparse functional regions, and a scarcity of available labels. These issues are further exacerbated in practice by the fact most real-life design scenarios necessitate the simultaneous optimization of multiple properties. In this work, we introduce ProteinNPT, a non-parametric transformer variant tailored to protein sequences and particularly suited to label-scarce and multi-task learning settings. We first focus on the supervised fitness prediction setting and develop several cross-validation schemes which support robust performance assessment. We subsequently reimplement prior top-performing baselines, introduce several extensions of these baselines by integrating diverse branches of the protein engineering literature, and demonstrate that ProteinNPT consistently outperforms all of them across a diverse set of protein property prediction tasks. Finally, we demonstrate the value of our approach for iterative protein design across extensive in silico Bayesian optimization and conditional sampling experiments.

## 1 Introduction

Proteins, vital to most biological processes, have evolved over billions of years, shaping their structure through countless evolutionary experiments. The vastness of the protein sequence space offers immense opportunities for enhancing the properties of existing proteins, paving the way for advancements in healthcare, sustainability, and the development of novel materials [Arnold, 2018, Huang et al., 2016]. Recent progress in sequencing technologies and machine learning methods bear the promise to explore this space more efficiently and design *functional* proteins with the desired properties. A key step toward that goal is the ability to predict the *fitness* of novel protein sequences – i.e., learning a mapping between structure and fitness of proteins, commonly referred to as a 'fitness landscape' [Romero et al., 2013]. However, the protein space is sparsely functional and experimentally-collected labels about properties of interest are scarce relative to the size of protein space [Biswas et al., 2021a]. Additionally, real-life protein design scenario often seek to optimize multiple properties of the protein simultaneously, which further aggravates the label scarcity issues. Historically, learning fitness landscapes has been tackled by two different lines of research in the computational biology literature. Firstly, supervised methods are trained to approximate the

---

[*]Correspondence: pascal.notin@cs.ox.ac.uk, ruben.weitzman@cs.ox.ac.uk; [†] Equal contribution

37th Conference on Neural Information Processing Systems (NeurIPS 2023).

fitness landscape by learning from experimental labels measuring a phenotype that is hypothesized to correlate with the function of interest. These methods have for instance been shown to lead to more efficient directed evolution pipelines [Dougherty and Arnold, 2009, Yang et al., 2019, Wittmann et al., 2020]. Given the scarcity of labels, they are typically constrained in terms of parameter count and expressivity of the underlying model (e.g., ridge regression, shallow neural network) [Heinzinger et al., 2019, Dallago et al., 2021, 2022, Stärk et al., 2021] to avoid overfitting. Secondly, unsupervised fitness predictors have reached increasingly higher fitness prediction performance without being subject to the limitations and potential biases stemming from training on sparse labels [Hopf et al., 2017a, Riesselman et al., 2018, Laine et al., 2019, Frazer et al., 2021, Meier et al., 2022, Notin et al., 2022a,b, Marquet et al., 2022]. Recently, Hsu et al. [2022] proposed to combine these two lines of research into a unified architecture in which a ridge regression is fed both one-hot encodings of input protein sequences and unsupervised fitness scores, and then trained on experimental labels to reach increased performance across several fitness prediction benchmarks. Yet, one-hot encoding features offer a relatively contrived and brittle sequence representation. For instance, they do not allow to make predictions at unseen positions resulting in low performance when attempting to extrapolate across positions (§ 4.3). Embeddings from protein language models offer a promising alternative but, since the resulting dimension of the full sequence embedding is typically too high, prior methods have resorted to dimensionality reduction techniques such as mean-pooling embeddings across the full sequence [Alley et al., 2019, Biswas et al., 2021a]. While this works reasonably well in practice, critical information may be lost in the pooling operation and, since not all residues may be relevant to a given task, we may want to be selective about which ones to consider. In this work, we introduce ProteinNPT (§ 3), a non-parametric transformer [Kossen et al., 2022] variant which is ideally suited to label-scarce settings through an additional regularizing denoising objective, straightforwardly extends to multi-task optimization settings and addresses all aforementioned issues. In order to quantify the ability of different models to extrapolate to unseen sequence positions, we devise several cross-validation schemes (§ 4.1) which we apply to all Deep Mutational Scanning (DMS) assays in the ProteinGym benchmarks [Notin et al., 2022a]. We then show that ProteinNPT achieves state-of-the-art performance across a wide range of protein property prediction and iterative redesign tasks. We summarize our contributions are as follows:

- We introduce ProteinNPT, a semi-supervised conditional pseudo-generative model for protein property prediction and design tasks (§ 3);

- We explore novel aspects of non-parametric transformers such as the use of auxiliary labels (§ 3.2), the multiple property prediction setting (§ 4.5), conditional sampling (§ 3.3) and uncertainty quantification (Appendix G.1);

- We devise several cross-validation schemes to assess the performance of fitness predictors in the supervised setting and their ability to extrapolate across positions (§ 4.1);

- We implement several of the current top-performing baselines, illustrate certain of their limitations, then improve these baselines accordingly by integrating ideas from diverse branches of the protein fitness prediction and engineering literature (§ 4.2);

- We subsequently show that ProteinNPT outperforms all of these baselines across a wide spectrum of property prediction tasks (§ 4.3-4.5);

- We demonstrate the potential of ProteinNPT for protein design through a diverse set of in silico Bayesian optimization (§ 5.1) and conditional sampling (§ 5.2) experiments.

## 2 Related work

**Protein fitness prediction**  Learning a fitness landscape is typically cast as a discriminative supervised learning task [Yang et al., 2018, Gelman et al., 2020, Freschlin et al., 2022]. Given the usually scarce number of labels relative to the dimensionality of the input space, the majority of approaches to date have relied on lightweight parametric models [Stärk et al., 2021, Heinzinger et al., 2019] of the primary protein structure and, occasionally, non-parametric models based on the tertiary protein structure [Romero et al., 2013]. Model features have revolved primarily around hand-crafted physico-chemical properties [Wittmann et al., 2020], one-hot-encodings of the amino acid sequence [Dallago et al., 2022] and, more recently, embeddings from deep generative models of evolutionary sequences [Alley et al., 2019, Brandes et al., 2021, Hesslow et al., 2022, Elnaggar et al., 2023a]. Deep generative models have also been successfully leveraged for zero-shot fitness

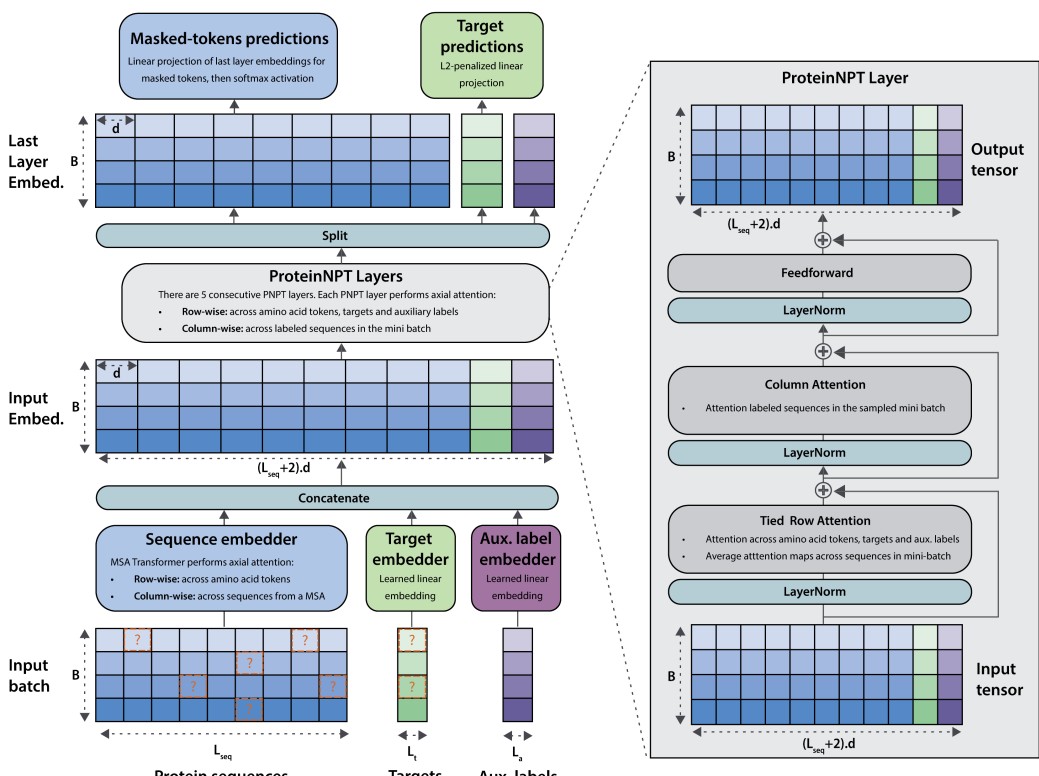

Figure 1: **ProteinNPT architecture.** (Left) The model takes as input the primary structure of a batch of proteins of length $L_{seq}$ along with the corresponding $L_t$ labels and, optionally, $L_a$ auxiliary labels (for simplicity we consider $L_t = L_a = 1$ here). Each input is embedded separately, then all resulting embeddings are concatenated into a single tensor. Several ProteinNPT layers are subsequently applied to learn a representation of the entire batch, which is ultimately used to predict both masked tokens and targets (depicted by question marks). (Right) A ProteinNPT layer alternates between tied row and column attention to learn rich embeddings of the labeled batch.

prediction [Hopf et al., 2017a, Riesselman et al., 2018, Meier et al., 2022, Notin et al., 2022a]. Their use is particularly compelling when experimental labels are not available or difficult to obtain for the property of interest. For instance, they enable accurate prediction of the effects of genetic mutations on human disease risks [Frazer et al., 2021, Brandes et al., 2022, Jagota et al., 2023], or can identify viral mutants with high escape potential at the beginning of a pandemic [Hie et al., 2020, Thadani et al., 2023].

**Protein redesign** The goal of protein redesign is to iteratively mutate a starting protein, typically a naturally occurring sequence, in order to enhance one or several of its properties, such as its stability, substrate specificity, or catalytic efficiency. Directed evolution [Arnold, 2018] is a widely-used experimental method in this field, emulating the way natural evolution works, but within a controlled lab environment. In each cycle, the starting protein sequences undergo random mutations, and the desired traits are then measured experimentally. The most promising variants are selected as the basis for the next round, and the process is repeated through several cycles until the desired design goals are achieved. Machine learning [Yang et al., 2019, Wu et al., 2019] has been shown to significantly enhance this process by exploring the combinatorial space of mutations more efficiently, for instance by reaching the same final performance with fewer experimental rounds [Biswas et al., 2021b].

**Self-attention over multi-dimensional objects** Popularized by the transformer architecture [Vaswani et al., 2017], self-attention enables to learn rich representations of complex structured objects, such as natural language sequences. One of its key limitations is the quadratic complexity of the attention mechanism with respect to the length of the input. Axial attention [Ho et al., 2019]

addresses that issue for multi-dimensional objects such as images by decomposing the traditional attention mechanism along each axis: separate self-attention mechanisms are successively applied on each axis, thereby drastically reducing computational complexity. The MSA Transformer [Rao et al., 2021] leverages axial attention to learn representations of a large and diverse set of Multiple Sequence Alignments (MSA) – two-dimensional objects comprised of homologous protein sequences aligned in the same coordinate system. As MSAs encapsulate rich evolutionary and structural information about the corresponding proteins, the MSA Transformer has lead to strong performance for several tasks, such as zero-shot contact prediction, fitness prediction [Meier et al., 2022] and protein engineering [Sgarbossa et al., 2023]. Viewing tabular data as two-dimensional objects, Non-Parametric Transformers [Kossen et al., 2022] applies axial attention to learn a representation of an entire labeled training dataset in a supervised setting. Self-attention is thus used to model relationships both between features and labels (across columns), and between labeled objects (across rows).

## 3 Methods

### 3.1 ProteinNPT model

**Model architecture** ProteinNPT (Fig. 1) is a semi-supervised conditional pseudo-generative model that learns a joint representation of protein sequences and associated property labels. The model takes as input *both* the primary structure representation of the proteins along with the corresponding labels for the property of interest. Let $(X^{\text{full}}, Y^{\text{full}})$ be the full training dataset where $X^{\text{full}} \in \{1, 20\}^{N.L_s}$ are protein sequences (with N the total number of labeled protein sequences and L the sequence length), and $Y^{\text{full}} \in \mathbb{R}^{N.T}$ the corresponding property labels (where $T$ is the number of distinct such labels, including $L_t$ true targets and $L_a$ auxiliary labels, as discussed in § 3.2). Depending on whether we are at training or inference time, we sample a batch of $B$ points and mask different parts of this combined input as per the procedure described later in this section. We separately embed protein sequences and labels, concatenate the resulting sequence and label embeddings (each of dimension $d$) into a single tensor $Z \in \mathbb{R}^{(B.(L_s+T).d)}$, which we then feed into several ProteinNPT layers. A ProteinNPT layer (Fig. 1 - right) learns joint representation of protein sequences and labels by applying successively self-attention between residues and labels for a given sequence (row-attention), self-attention across sequences in the input batch at a given position (column-attention), and a feedforward layer. Each of these transforms is preceded by a LayerNorm operator $LN(.)$ and we add residual connections to the output of each step. For the multi-head row-attention sub-layer, we linearly project embeddings for each labeled sequence $n \in \{1, B\}$ for each attention head $i \in \{1, H\}$ via the linear embeddings $Wr_i^K$, $Wr_i^Q$ and $Wr_i^V$ respectively. Mathematically, we have:

$$\text{Row-Att}(Z) = Z + \text{Tied-Row-MHSA}(LN(Z)) = Z + \text{concat}(O_1, O_2, ..., O_H).W^O \quad (1)$$

where the concatenation is performed row-wise, $W_O$ mixes outputs $O_i$ from different heads, and we use tied row-attention as defined in Rao et al. [2021] as the attention maps ought to be similar across labeled instances from the same protein family:

$$\text{Tied-Row-Att}((Q_n, K_n, V_n)_{n \in \{1, B\}}) = \text{softmax}(\sum_{n=1}^{B} \frac{Q_n.K_n^T}{\sqrt{B.d}}).V_n \quad (2)$$

We then apply column-attention as follows:

$$\text{Col-Att}(Z) = Z + \text{MHSA}(LN(Z)) = Z + \text{concat}(P_1, P_2, ..., P_H).W^P \quad (3)$$

where the concatenation is performed column-wise, $W_P$ mixes outputs $P_i$ from different heads, and the standard self-attention operator $\text{Att}(Q, K, V) = \text{softmax}(Q.K^T/\sqrt{d}).V$. Lastly, the feedforward sub-layer applies a row-wise feed-forward network:

$$\text{FF}(Z) = Z + \text{rFF}(LN(Z)) \quad (4)$$

In the final stage, the learned embeddings from the last layer are used to predict both the masked tokens and targets: the embeddings of masked targets are input into a L2-penalized linear projection to predict masked target values, and the embeddings of masked tokens are linearly projected then input into a softmax activation to predict the corresponding original tokens.

**Model training**   Our overall architecture is trained with a two-stage semi-supervised procedure. First, due to the scarcity of labeled instances relative to the size of the underlying protein sequence space, we leverage embeddings from protein language models that have been pre-trained on large quantities of *unlabeled* natural sequences. Throughout training, the parameters of the model used to obtain these protein embeddings are frozen to prevent overfitting. This also helps alleviating several practical challenges pertaining to GPU memory bottlenecks at train time (Appendix C.3). Second, similarly to what was proposed in Non-Parametric Transformers [Kossen et al., 2022], we train our architecture on a composite task combining input denoising and target prediction. In our case, the former is equivalent to a masked language modeling objective on amino acid tokens [Devlin et al., 2019]. We randomly mask a fraction of amino acids and of input labels across all sequences in the mini batch (15% for both), and seek to predict these masked items. Our training loss is thus comprised of two components: 1) a reconstruction loss over masked amino acids 2) a target prediction loss over masked targets:

$$\mathcal{L}^{\text{total}} = \alpha_t . \mathcal{L}^{\text{AA reconstruction}} + (1 - \alpha_t) . \mathcal{L}^{\text{target prediction}} \tag{5}$$

where $\mathcal{L}^{\text{target prediction}}$ is target prediction loss (e.g., Mean Squared Error between predictions for the *masked* labels and corresponding true labels for continuous targets), $\mathcal{L}^{\text{AA reconstruction}}$ the cross-entropy loss for the masked input tokens, and $\alpha_t$ balances out the two objectives. As suggested by Kossen et al. [2022], $\alpha_t$ is progressively annealed to increasingly focus on target prediction during training. The amino acid denoising objective entices the network to learn a representation of the full dataset, which in turn acts as a regularization mechanism which is beneficial in label scarce settings.

**Inference**   At inference, we form input batches by concatenating row-wise a fraction of test sequences (with unknown, masked labels) with training instances (with known, unmasked labels). Predictions are therefore dependent on both the learned relationships between tokens and targets, as well as homology across labeled instances. When the size of the training set is large enough, we cannot use the entire training set at inference due to memory bottlenecks, and have to resort to sampling training instances. We explore the impact of the number of training sequences sampled at inference in § 6.

**Advantages of tri-axial attention**   To obtain the initial protein sequence embeddings, we experimented with several best-in-class protein language models and obtained better results with the MSA Transformer (Appendix C.2). As a result, our overall architecture operates *tri-axial* self-attention to output predictions: across residues and labels, across homologous sequences from the retrieved MSA (in MSA Transformer layers only) and across labeled examples (in ProteinNPT layers only). There are several advantages conferred by this architecture. First, it seamlessly adapts to multiple target predictions, by concatenating as many label columns in its input as required. Second, it naturally captures correlations between these targets and automatically handles – at training and at inference – observations with partially available labels. Third, to avoid overfitting to sparse label sets, prior semi-supervised approaches typically apply a pooling operator (e.g., average pooling across the full sequence length) to reduce the dimensionality of input features. This pooling operation potentially destroys valuable information for the downstream task. In contrast, no such pooling operation is applied in the ProteinNPT architecture, and the network leverages self-attention to learn dependencies between labels and the embeddings of specific residues in the sequence.

## 3.2   Auxiliary labels

The effectiveness of our architecture is significantly enhanced by the incorporation of *auxiliary labels* at training and inference. We define auxiliary labels as additional inputs which are both 1) easy to obtain for all or for a subset of train and test instances 2) known to be correlated with the target of interest. They are concatenated column-wise to the input batch and are handled by ProteinNPT layers in the forward pass just like any other target. However, they are excluded from the loss computation and are only meant to increase performance on the target prediction task by instilling complementary information into the model. In the fitness prediction tasks discussed in § 4, we use as auxiliary labels the *unsupervised fitness predictions* obtained with the underlying protein language model used to extract the input sequence embeddings (i.e., MSA Transformer). As shown in ablations (Appendix C.2), and consistent with the findings from Hsu et al. [2022], this helps significantly increase our prediction performance, in particular when performing inference at positions where no mutation was observed during training.

### 3.3 Conditional sampling

Diverging from prior works focusing on supervised protein property predictions (§ 2 and § 4.2), ProteinNPT learns a joint representation of both protein sequences and labels. Since it is trained with a masked-language model objective, its output at a given masked position is a distribution over the amino acid vocabulary at that position conditioned on non-masked tokens, including labels. As such, it is a conditional *pseudo-generative* model which can be used to sample new sequences iteratively. Prior works have investigated sampling with protein masked-language models in the unsupervised setting [Sgarbossa et al., 2023]. However, a distinctive feature of our architecture is its ability to condition on values of the property of interest. This approach allows to steer the generation of new sequences to align with specific protein design goals, as detailed in § 5.2.

## 4 Protein property prediction

### 4.1 ProteinGym benchmark

The fitness landscape of naturally occurring proteins is the result of an intricate set of overlapping constraints that these proteins are subjected to in an organism. It is thus challenging to identify a single molecular property that is both easy to measure experimentally and that reflects that complexity. Experimental measurements are imperfect ground truth measurements of the underlying fitness. Furthermore, there is no such thing as fitness in the absolute but rather fitness at a specific temperature, pH, or in a given cell type. To make the conclusions that we seek to draw in the subsequent sections more meaningful, it is critical to consider a large and diverse set of experimental assays. To that end, we conducted our experiments on ProteinGym [Notin et al., 2022a], which contains an extensive set of Deep Mutational Scanning (DMS) assays covering a wide range of functional properties (e.g., thermostability, ligand binding, viral replication, drug resistance). We include an additional 13 DMS assays to ProteinGym, bringing the total to 100 DMS assays (Appendix A).

We develop 3 distinct cross-validation schemes to assess the ability of each model to extrapolate to positions not encountered during training. In the *Random* scheme, commonly-used in other supervised fitness prediction benchmarks [Rao et al., 2019, Dallago et al., 2022], each mutation is randomly allocated to one of five distinct folds. In the *Contiguous* scheme, the sequence is split into five contiguous segments along its length, with mutations assigned to each segment based on the position they occur in the sequence. Lastly, the *Modulo* scheme uses the modulo operator to assign mutated positions to each fold. For example, position 1 is assigned to fold 1, position 2 to fold 2, and so on, looping back to fold 1 at position 6. This pattern continues throughout the sequence. We note that there is no inherent issue with using a Random cross-validation scheme to estimate the performance of predictive models. However, the conclusions drawn and the generalizability claims based on it require careful consideration.

### 4.2 Baselines

We start from the top-performing model variant from [Hsu et al., 2022], namely a ridge regression on one-hot encodings of protein sequences, augmented with unsupervised fitness predictions from DeepSequence [Riesselman et al., 2018]. Building on recent progress from the protein property prediction and engineering literature, we then improve this baseline in several ways. First, we show that augmenting the model from Hsu et al. [2022] with the latest state-of-the-art unsupervised fitness predictors helps further boost the predictive performance, with the performance boost being consistent with the zero-shot performance of the corresponding unsupervised models (Appendix D). Then, considering the performance across the various cross-validation schemes described in § 4.1, we show the limits of One-Hot-Encoded (OHE) features to generalize across positions and propose to use instead embeddings from large protein language models. There is a rich literature investigating this idea, and we test out a wide range of model architectures that have been proposed to learn a mapping between embeddings and labels: L2-penalized regression, shallow Multi-Layer Perceptron (MLP), Convolutional Neural Network (CNN), ConvBERT [Elnaggar et al., 2023b] and Light Attention [Stärk et al., 2021]. Combining these different ideas together, we obtain the main baselines we compare against in subsequent sections: models that both leverage embeddings from pretrained language models, and are augmented with unsupervised fitness predictions. We carry out thorough ablations and hyperparameter search (Appendix C.2) to identify the best performing combinations.

## 4.3 Single mutant property prediction

Table 1: **Fitness prediction performance**. Spearman's rank correlation & MSE between model predictions & experimental measurements, averaged across all assays in ProteinGym. DS and MSAT are shorthand for DeepSequence and MSA Transformer respectively. "Aug." indicates models augmented with unsupervised predictions. The Spearman for MSAT in the zero-shot setting is 0.41 (independent of CV split). Assay-level performance & standard errors are reported in Appendix D.

| Model name | Spearman ($\uparrow$) | | | | MSE ($\downarrow$) | | | |
|---|---|---|---|---|---|---|---|---|
| | Contig. | Mod. | Rand. | Avg. | Contig. | Mod. | Rand. | Avg. |
| OHE | 0.08 | 0.02 | 0.54 | 0.21 | 1.17 | 1.11 | 0.92 | 1.06 |
| OHE - Aug. (DS) | 0.40 | 0.39 | 0.48 | 0.42 | 0.98 | 0.93 | 0.78 | 0.90 |
| OHE - Aug. (MSAT) | 0.41 | 0.40 | 0.50 | 0.44 | 0.97 | 0.92 | 0.77 | 0.89 |
| Embed. - Aug. (MSAT) | 0.47 | 0.49 | 0.57 | 0.51 | **0.93** | 0.85 | 0.67 | 0.82 |
| ProteinNPT | **0.48** | **0.51** | **0.65** | **0.54** | **0.93** | **0.83** | **0.53** | **0.77** |

**Experimental setup** We first focus on the task of predicting fitness for single substitution mutants. For each assay in ProteinGym and cross-validation scheme introduced in § 4.1, we perform a 5-fold cross-validation, selecting the first 4 folds for training, and using the remaining one as test set. Intuitively, since the Random cross-validation scheme is such that mutations in the test set may occur at the same positions as certain mutations observed in the training data, model performance will typically be higher in that regime. Similarly, the Contiguous scheme is intuitively the most difficult regime in which we assess the ability of the different models to extrapolate on positions relatively 'far' in the linear sequence from any mutation being part of the training data.

**Results** We find that ProteinNPT markedly outperforms all baselines, across all cross-validation schemes, and both in terms of Spearman's rank correlation and Mean Squared Error (MSE) (Table 1). Our results also confirm that a model that would learn solely based on one-hot-encoded inputs would be unable to make any valuable prediction at positions that were not observed at training time (Spearman's rank correlation near zero on the Contiguous and Modulo schemes). Augmenting these models with predictions from unsupervised predictors helps to generally address that issue, although the performance on the Contiguous and Modulo schemes is similar to the zero-shot performance of unsupervised models (the average Spearman's rank correlation across all assays for the MSA Transformer in the zero-shot setting is 0.41). The baseline combining a transform of embeddings from the MSA Transformer, augmented with unsupervised fitness predictions ('Embeddings - Augmented (MSAT) in Table 1) performs better across the board than the one-hot encoding equivalent, yet its performance is far from that of ProteinNPT, in particular in the Random cross-validation scheme. We interpret the latter by the fact the column-wise attention model in the ProteinNPT layer (Fig. 1) is more beneficial to performance when a mutation at that position has been observed at train time.

## 4.4 Multiple mutants property prediction

**Experimental setup** We next consider all assays in the extended ProteinGym benchmark that include multiple mutants. Given the challenges to assign multiple mutants to non-overlapping position sets, we only consider the Random cross-validation scheme in this analysis. We report the Spearman's rank correlation in Fig. 2 and MSE in Appendix E (conclusions are consistent for the two metrics).

**Results** The performance lift of ProteinNPT observed in the single mutants analyses translates to multiple mutants: ProteinNPT has superior aggregate performance and outperforms all other baselines in 14 out of the 17 assays with multiple mutants (Fig. 2, left). The performance lift is also robust to the mutational depth considered (Fig. 2, right).

## 4.5 Multiple property prediction

ProteinGym contains a subset of protein families for which several experimental assays have been carried out, offering the possibility to test out the ability of various models to predict multiple properties of interest simultaneously. We report the performance on the Random cross-validation scheme in Fig. 9, and other schemes are provided in Appendix E. ProteinNPT outperforms all other baselines in that regime as well, consistently across all settings analyzed.

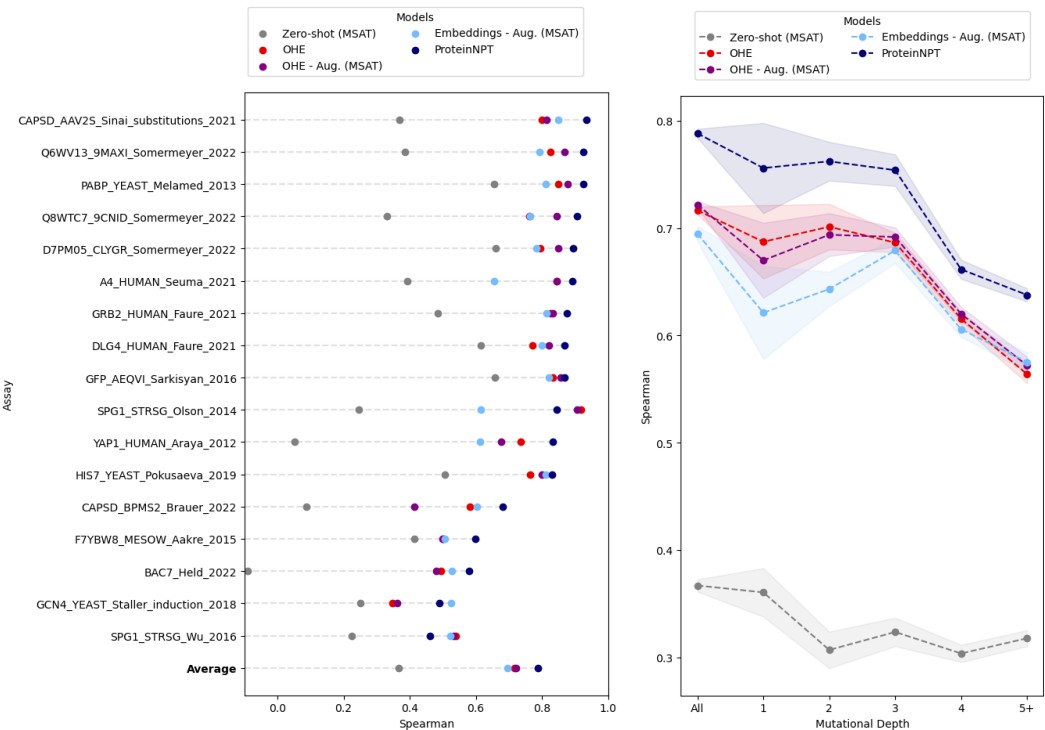

Figure 2: **Multiples mutants performance.** (Left) Spearman's rank correlation between model predictions and experimental measurements, for each assay in ProteinGym with multiple mutants (see Appendix A.1). (Right) Average Spearman's rank correlation, overall and by mutational depth.

## 5 Protein redesign

### 5.1 In silico iterative protein redesign

**Experimental setup** Our goal is to emulate *in silico* a real-world protein engineering scenario where we iteratively redesign a known protein sequence across several experimental rounds to increase its fitness. Performing multiple rounds of experiments sequentially allows us to harness the information acquired in prior rounds, as is commonly done in directed evolution. In a standard wet lab setting, there is no constraint on the subset of mutated sequences that can be experimentally tested. However, in our in silico setup, we are constrained to select mutants that belong to a predefined pool of sequences which have been experimentally measured in the DMS assays we consider, such as the ones from ProteinGym. We provide a detailed algorithm for our procedure in Fig. 3 (left), which we cast as a batch Bayesian optimization task. At the beginning of each in silico experiment, this pool is entirely unlabeled. In each subsequent round, we score all possible variants in the unlabeled pool and select from it a subset $B$ of items which are deemed optimal by the Upper Confidence Bound (UCB) acquisition function:

$$a(x; \lambda) = \mu(x) + \lambda.\sigma(x) \tag{6}$$

where $\mu$ is the predicted fitness value, $\sigma$ quantifies uncertainty in the prediction, and $\lambda$ controls the exploration-exploitation trade-off. We develop and test various uncertainty quantification methods for non-parametric transformer architectures, based on Monte Carlo dropout [Gal and Ghahramani, 2015] and resampling inference batches with replacement (see details in Appendix G.1). We find that a *hybrid* scheme in which we both sample model parameters via Monte Carlo dropout and resample inference batches works best. After each acquisition cycle, all models are retrained from scratch for 2k training steps. On Fig. 3, we plot the proportion of acquired points that are above a predefined threshold (top 3 deciles of all measurements in the DMS assays) as a function of the number of acquisition rounds performed, averaged across all assays. The baselines considered are the same as for the property prediction experiments. Individual plots for each assay are provided in Appendix G.3.

**Results**   We find that ProteinNPT vastly outperforms all other baselines in aggregate across all assays, and achieves superior performance on the large majority of individual assays. Furthermore, we observe that the performance lift is more pronounced for assays for which we start with a larger pool of unlabeled sequences. This suggests that in a real-world redesign scenario, in which choices are not limited to a predefined pool as in our in silico setting, ProteinNPT might exhibit even greater performance gains relative to the baseline models.

**Algorithm 1 Iterative protein redesign**

1: **Input:** Initial labeled data $\mathcal{D}_L$; Initial unlabeled data $\mathcal{D}_U$; Batch size $B$; Batch set $S = \emptyset$; Acquisition function $\alpha(\boldsymbol{x}; \lambda)$
2: **for** $t \in 1, 2, \ldots, 10$ **do**
3:   Train model model on $\mathcal{D}_L$
4:   **for** $t \in 1, 2, \ldots, B$ **do**
5:     $\boldsymbol{x}_{\text{new}} = \arg\max_{\boldsymbol{x} \in \mathcal{D}_U} \alpha(\boldsymbol{x}; \lambda)$
6:     Obtain label $y_{\text{new}}$ for $\boldsymbol{x}_{\text{new}}$
7:     $S \leftarrow S \cup \{(\boldsymbol{x}_{\text{new}}, y_{\text{new}})\}$
8:     $\mathcal{D}_U \leftarrow \mathcal{D}_U \setminus \{\boldsymbol{x}_{\text{new}}\}$
9:   **end for**
10:   $\mathcal{D}_L \leftarrow \mathcal{D}_L \cup S, S \leftarrow \emptyset$
11: **end for**
12: **Output:** $\mathcal{D}_L$, Trained model on $\mathcal{D}_L$

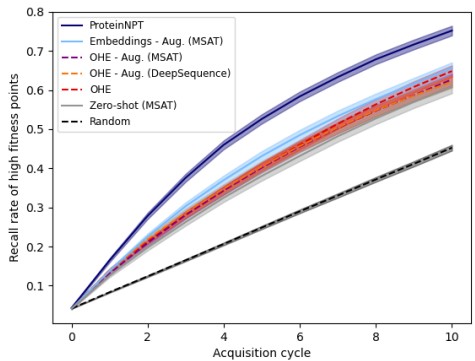

Figure 3: **In silico protein redesign.** (Left) Iterative redesign algorithm (Right) Recall rate of high fitness points (top 3 deciles) vs acquisition cycle, averaged across all DMS assays in ProteinGym.

## 5.2   Conditional sampling

As a supervised pseudo-generative model of labeled protein sequences, ProteinNPT has the ability to generate novel sequences conditioned on particular values of these properties. This is in contrast to prior generative models that have been used to sample natural-like proteins with no bias towards properties of interest [Repecka et al., 2021, Sgarbossa et al., 2023], or conditioned on broad taxonomic labels characterizing full protein families [Madani et al., 2023] as opposed to the sequence-specific properties we consider here. In this experiment, we illustrate the ability of ProteinNPT to sample novel protein sequences with desired property from the Green Fluorescent Protein (GFP), but the approach can be applied to any other protein. We first identify the sequence with the highest measured property in the GFP DMS assay [Sarkisyan et al., 2016]. We then form an input batch by randomly selecting other labeled sequences, mask a fixed number of tokens (5 in our experiment) in the fittest sequence, obtain the resulting log softmax over the masked positions with a forward pass and sample from these to obtain new sequences. Critically, rather than selecting the masked positions at random, we sample them from the positions with the highest row-wise attention coefficient with the target (averaged across heads) in the last layer of ProteinNPT. This helps ensure we sample new mutations at the positions with the highest impact on the target of interest. We generate 1k new sequences with that process and measure the corresponding fitness with an independent zero-shot fitness predictor (ESM-1v). Finally, we compare the resulting fitness distribution across all samples with distributions obtained with two baselines: the first in which mutations are randomly selected, and the second in which mutations are selected to further minimize the fitness of the less fit protein. We report the corresponding results in Figure 4, and observe that proteins sampled with ProteinNPT have substantially higher fitness values relative to the two baseline sampling strategies and relative to sequences in the original DMS assay.

## 6   Discussion

**Row-wise attention captures dependencies between residues and function.**   Since we learn a joint representation of sequence and labels, the attention maps of ProteinNPT capture meaningful correlations between residues and the property labels we train on. We represent the row-wise attention coefficients for a ProteinNPT model trained on a DHFR protein assay [Thompson et al., 2020a] in Fig. 5 (last layer, first attention head). The analysis recapitulates certain known facts about important residues for the assayed property (e.g., the highest attention score in red corresponds to a core substrate binding site), and may help identify new residues involved the protein function of interest.

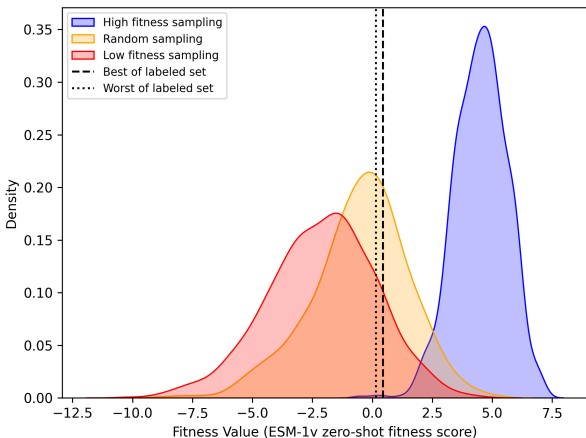

Figure 4: **Conditional sampling.** ProteinNPT is used to sample novel sequences for the GFP protein conditioned on high fitness values, leading to sequences with high predicted fitness relative to controls.

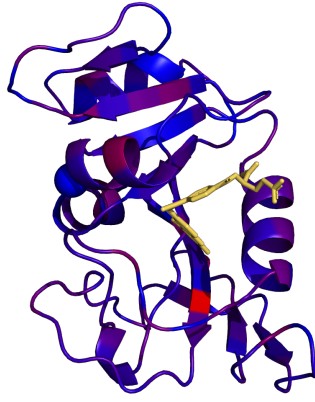

Figure 5: **Row-wise attention map.** Row-wise attention between residues & fitness, mapped on the DHFR enzyme structure. The high intensity residue (red) corresponds to a substrate binding site.

Table 2: **Impact of column-wise attention on performance.** (Left) Impact of ablating the column-wise attention in ProteinNPT layers. (Right) Impact of the number of labeled sequences used at inference for a model trained with column-wise attention. Performance is measured by the Spearman's rank correlation between model predictions and experimental measurements.

| CV scheme | Column attention | | Nb. of labeled sequences at inference | | | | | |
|---|---|---|---|---|---|---|---|---|
| | Without | With | 0 | 100 | 200 | 500 | 1000 | 2000 |
| Random | 0.669 | **0.684** | 0.398 | 0.677 | 0.678 | 0.679 | 0.684 | **0.685** |
| Modulo | 0.530 | **0.531** | 0.299 | **0.533** | 0.531 | 0.531 | 0.531 | 0.531 |
| Contiguous | 0.425 | **0.501** | 0.254 | 0.496 | **0.504** | 0.502 | 0.501 | 0.500 |
| Average | 0.542 | **0.572** | 0.317 | 0.569 | 0.571 | 0.571 | **0.572** | 0.572 |

**Column-wise attention is critical to reach top performance.** We test out the impact of column-attention on model performance. All analyses are conducted on the same subset of 8 DMS assays used in other ablations (Appendix B.2). We first re-train ProteinNPT models without the column-wise attention layers, and observe that the downstream performance drops significantly (Table 2, left). At inference, we find that predicting with 100 randomly-selected labeled sequences captures most of the performance lift, and observe no benefit beyond 1k batch points (Table 2, right).

# 7 Conclusion

This work introduced ProteinNPT, a novel conditional pseudo-generative architecture which reaches state-of-the-art performance across a wide range of fitness prediction and iterative redesign settings, covering a diverse set of protein families and assayed properties (e.g, thermostability, binding, enzymatic activity). The architecture straightforwardly extends to the multiple property prediction setting, can leverage auxiliary labels, supports uncertainty quantification, and allows to sample new sequences conditioned on desired property values. While sequence embeddings from the MSA Transformer lead to the best performance in our ablations, the architecture of ProteinNPT is agnostic to the model used to obtain these embeddings. As protein language models continue to improve, ProteinNPT will also benefit from the resulting improved embeddings. Future work will investigate additional strategies for conditional sampling, extension of the architecture to handle insertions and deletions, and generalization across protein families.

## Acknowledgements

P.N. is supported by GSK and the UK Engineering and Physical Sciences Research Council (ESPRC ICASE award no.18000077). R.W. is supported by the UK Engineering and Physical Sciences Research Council, as part of the CDT in Health Data Science. D.S.M. is supported by a Chan Zuckerberg Initiative Award (Neurodegeneration Challenge Network, CZI2018-191853) and a NIH Transformational Research Award (TR01 1R01CA260415). Y.G. holds a Turing AI Fellowship (Phase 1) at the Alan Turing Institute, which is supported by EPSRC grant reference V030302/1. We thank Jannik Kossen for helpful feedback and discussion.

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

# Appendix

## A    ProteinGym extensions

### A.1    Curation of additional DMS assays

In this work we extensively leverage the ProteinGym benchmark introduced in Notin et al. [2022a]. We further expand the available pool of DMS assays by incorporating 13 additional assays, resulting in a total of 100 DMS assays. We refer to this broader pool in the text as 'the extended benchmark'. The selection of these assays was guided by various criteria, including recency (we included recent DMS experiments published in 2022 and 2023, not available when ProteinGym was initially released) and quality (e.g., high number of mutants, high correlation between replicates). We also favored the inclusion of assays that contain multiple mutants, as it was one of the shortcomings of the first ProteinGym benchmarks. Lastly, we focused on DMS assays that involved proteins relevant to protein design, such as GFP, capsid proteins, antitoxins, antimicrobial agents, CRISPR/Cas9, and kinases. We give further details on all assays that belong to the extended benchmark in the reference file included in the code repository. The 13 new assays that we include are shown in Table 3.

Table 3: DMS assays added to ProteinGym to for the extended benchmark

| Uniprot ID | Reference | Selection assay | Function grouping |
|---|---|---|---|
| Q8WTC7 9CNID | Gonzalez Somermeyer et al. [2022] | Green fluorescence intensity | Activity |
| D7PM05 CLYGR | Gonzalez Somermeyer et al. [2022] | Green fluorescence intensity | Activity |
| Q6WV13 9MAXI | Gonzalez Somermeyer et al. [2022] | Green fluorescence intensity | Activity |
| SPG1 STRSG | Wu et al. [2016a] | Binding affinity to IgG | Binding |
| CAPSD BPMS2 | Brauer et al. [2021] | Quantitative capsid self-assembly | Activity |
| PARE ECOLX | Ding et al. [2022] | Growth under toxin | Activity |
| CAS9 STRP1 | Spencer and Zhang [2017] | Expression of Cas9 and correct cleavage | Activity |
| CAS9 STRP1 | Spencer and Zhang [2017] | Expression of Cas9 and non-cleavage of off target sequence | Activity |
| CTHL3 BOVIN | Koch et al. [2022] | Antimicrobial activity in minimal inhibitory concentration | Activity |
| A0A247D711 LISMN | Stadelmann et al. [2021] | Activity against SpyCas9 inducing an RFP reporter | Activity |
| A0A220GHA5 9CAUD | Stadelmann et al. [2021] | Activity against SpyCas9 inducing an RFP reporter | Activity |
| HXK4 HUMAN | Gersing et al. [2022] | Functional complementation to reduced growth on glucose medium | Organismal Fitness |
| ANCSZ | Hobbs et al. [2022] | Successful phosphorylation of bait peptide | Activity |

For consistency, we pre-processed the raw DMS data following the approach from Notin et al. [2022a]. As such, we exclude silent mutations from our benchmark, which are nucleotide substitutions that do not affect the resulting protein sequence. Duplicate mutants, either arising from repeated nucleotide substitutions or indels leading to identical protein sequences, are removed by calculating the average of all DMS measurements across duplicates. Finally, mutants with missing assay measurements are excluded.

### A.2    Creation of cross-validation schemes

As described in the text, we designed 3 types of cross-validation schemes: Random, Contiguous and Modulo. For the Random scheme, we randomly assign each mutation to one of 5 folds, which may result in mutations at the same position to be assigned to different folds. The Contiguous scheme is obtained by splitting the sequence in contiguous segments along its length. We ensure that the segments are comprised of the same number of positions by first counting the total number of mutated position in the sequence. If this total is not divisible by the total number of fold, the first few folds will include one more position than the rest. We only consider positions mutated, which may not span the entire length of the protein sequence. 'Contiguous' therefore means contiguous only with respect to the mutated range. Lastly, the Modulo scheme is obtained by assigning positions to each fold using the modulo of the residue position. Therefore, for a 5 fold cross-validation, position 1 is

assigned to fold 1, position 2 to fold 2, ..., position 6 to fold 1, etc. Once again, we make sure to only consider mutated positions. If position 12 is assigned to fold 2, and position 13 and 14 are not mutated, position 15 will be assigned to fold 3. We operate a five fold cross-validation for all assays except for assays F7YBW8 MESOW Aakre et al. [2015] and SPG1 STRSG Wu et al. [2016b], as these contain only 4 mutated positions. We therefore do a 4-fold cross-validation for these two proteins (the Contiguous and Modulo schemes are therefore identical for these two proteins). Note that the Contiguous and Modulo cross-validation schemes are only used when assessing the effects of single mutants. Multiple mutants generally involve several positions that may not be easily separated into the independent folds as discussed in the Contiguous and Modulo schemes above. Thus, all analyses about multiple mutants conducted in this work (for example in § 4.4) are based on the Random cross-validation scheme only.

# B  ProteinNPT details

## B.1  Model architecture

Instead of learning a parametric model that makes protein property predictions based on the features of a single input protein sequence, our model makes predictions both based on these features but also based on similarity with a mini-batch of already labeled instances via self-attention. Using similarity between protein sequences is a cornerstone of computational biology and protein modeling via the use of Multiple Sequence Alignments (MSAs). The MSA Transformer showed how self-attention across the sequences of an MSA is a very effective way to learn compact protein representations in an unsupervised setting, and the idea of performing self-attention across labeled instances has first been proposed by Kossen et al. [2022] – we discuss how ProteinNPT relates to these two works in more details in Appendix B.5.

We summarize our architecture in Fig. 1. We embed the input mini-batch of protein sequences with a pre-trained protein language models. We experimented with various models in our ablations (Appendix C.2), obtained superior performance with the MSA Transformer and used this model in all subsequent analyses discussed in this work. Assuming the input sequence is of length $L$, the resulting embedded mini-bath of $B$ sequences is of size $B.L.D$, where $D$ is the dimension for the embedding of a single amino acid token [2]. To prevent overfitting when training a ProteinNPT model on a relatively small labeled dataset, the embedding dimension $d$ used in subsequent self-attention layers of the network is smaller than $D$ (e.g., $D = 768$ for the MSA Transformer, and $d = 200$ in our final architecture based on ablations). We apply linear projections to reduce dimensionality accordingly.

In what follows, we assume the input target labels and auxiliary labels are both unidimensional continuous inputs, but everything seamlessly extends to continuous or categorical inputs of arbitrary dimensions (our codebase handles both input types). In our final ProteinNPT architecture the auxiliary labels are zero-shot fitness predictions obtained with the MSA Transformer, following the procedure described in Meier et al. [2022]. To disambiguate masked entries from true input zeros, we append to continuous vectors a binary flag indicating whether the corresponding label is masked (for categorical input, a separate category corresponds to masked targets). After applying standard scaling on the input targets, we use linear layers (a separate one for each target and each auxiliary label) to project them into a $d$-dimensional vector. We then concatenate the protein sequence embeddings with the input target embeddings and auxiliary label embeddings.

We also experiment with various transforms of the resulting embeddings: Convolutional layer (CNN), ConvBERT layer [Jiang et al., 2021], or no transform. We find in our ablations that applying a CNN layer delivers the best performance on downstream tasks performance.

The transformed embeddings are subsequently fed into 5 consecutive ProteinNPT layers (chosen based on ablations). Each ProteinNPT layer applies successively row-attention, column-attention and a feedforward layer. Each of these transforms is preceded by a LayerNorm operator and we add residual connections to the output of each step. As proposed in the MSA Transformer, we leverage tied row-wise attention, ie., the attention maps in row attention are averaged across rows, resulting in lower memory footprint. This is a sensible design for the applications studied in our work, in which labeled sequences are relatively similar to one another and share a common three-dimensional contact structure.

Finally, we make predictions for the targets of interests by feeding the corresponding target embeddings into a L2-penalized linear projector (a separate linear map is learned for each target). We summarize all architecture details from our final architecture in Table 4.

---

[2]Depending on the underlying pre-trained protein language models used, additional beginning of sequence (BOS) and end of sequence (EOS) tokens are also added in practice. For instance, in the case of the MSA Transformer, a BOS token is prepended to the beginning of the input sequence. The size of the embedded sequence is thus $B.(L + 1).D$

Table 4: **ProteinNPT - Architecture details**

| Hyperparameter | Value |
|---|---|
| Nb. ProteinNPT layers | 5 |
| Embedding dimension ($d$) | 200 |
| Feedforward embedding dimension | 400 |
| Nb. attention heads | 4 |
| CNN kernel size (post embedding) | 7 |
| Weight decay | $5.10^{-3}$ |
| Dropout | 0.0 |

Table 5: **ProteinNPT - Training details**

| Hyperparameter | Value |
|---|---|
| Training steps | 10k |
| Learning rate warmup steps | 100 |
| Peak learning rate | $3.10^{-4}$ |
| Optimizer | AdamW |
| Gradient clipping norm | 1.0 |
| Learning rate schedule | Cosine |
| Training batch (masked) | 64 |
| Training batch (unmasked) | 361 |

Table 6: **ProteinNPT ablations - Use of auxiliary labels** Spearman's rank correlation between model predictions and experimental measurements. The auxiliary label used here is the zero-shot fitness prediction with the MSA Transformer.

| CV scheme | No aux. label | With aux. label |
|---|---|---|
| Random | 0.683 | **0.684** |
| Modulo | 0.527 | **0.531** |
| Contiguous | 0.439 | **0.501** |
| Average | 0.549 | **0.572** |

## B.2 Ablations

We carried out thorough ablations to develop ProteinNPT. Following [Meier et al., 2022, Notin et al., 2022a], to support theses various ablations, we set aside a relatively small set of DMS assays (8 assays out of the 100 assays in the extended ProteinGym substitution benchmark), which are used to decide between different model architectures while not overfitting these decisions to the benchmark:

- BLAT ECOLX [Jacquier et al., 2013]
- CALM1 HUMAN [Weile et al., 2017]
- DYR ECOLI [Thompson et al., 2020b]
- DLG4 RAT [McLaughlin Jr et al., 2012]

- P53 HUMAN [Giacomelli et al., 2018]
- REV HV1H2 [Fernandes et al., 2016]
- RL401 YEAST [Roscoe et al., 2013]
- TAT HV1BR [Fernandes et al., 2016]

Together, these 8 assays cover diverse protein families in terms of taxa, mutation depths, sequence lengths and MSA depth.

We report results for several of these ablations in this subsection, in particular varying the model used to obtain protein sequence embeddings (Table 7) and the use of auxiliary labels (Table 6). Based on these analyses, we use for our final model architecture the MSA Transformer [Rao et al., 2021] to obtain protein sequence embeddings, use auxiliary labels (ie. zero-shot fitness predictions with MSA Transformer), and use batches with 1k training instances to make predictions at inference.

Table 7: **ProteinNPT ablations - Protein sequence embeddings** Spearman's rank correlation between model predictions and experimental measurements, when leveraging different protein language models (ESM-1v, Tranception or MSA Transformer) to obtain sequence embeddings and using as auxiliary labels the zero-shot predictions from the corresponding protein language model. 'Learned' corresponds to the case of learned token embeddings from scratch and no auxiliary labels.

| CV scheme | Learned | Tranception | ESM1v | MSAT |
|---|---|---|---|---|
| Random | 0.422 | 0.665 | 0.659 | **0.684** |
| Modulo | 0.098 | 0.433 | 0.468 | **0.531** |
| Contiguous | -0.048 | 0.375 | 0.380 | **0.501** |
| Average | 0.141 | 0.491 | 0.502 | **0.572** |

Table 8: **ProteinNPT ablations - Impact of the size of the training set** Average Spearman's rank correlation between predictions and DMS labels. Assays are grouped based on 'Small size'($\leq$2k labels), 'Medium size' (2-8k labels) and 'Large size' (>8k labels).

| CV scheme | Small | Medium | Large |
|-----------|-------|--------|-------|
| Random | 0.633 | 0.738 | 0.573 |
| Modulo | 0.492 | 0.591 | 0.421 |
| Contiguous | 0.481 | 0.546 | 0.381 |
| Average | 0.535 | 0.625 | 0.458 |

## B.3 Model training

At train time, we feed as input to our model a batch of 425 protein sequences and the corresponding labels. Since we mask at random 15% of labels, this equates to having on average 64 masked training instances – which we want to accurately predict the values of, and 361 unmasked instances – which are used to support the predictions of masked instances. We also mask at random 15% of input protein sequence tokens. As discussed in § 3, our training loss is the weighted average of two different losses: the Cross-Entropy (CE) loss over masked tokens predictions (akin to a standard masked-language modeling objective [Devlin et al., 2019]) and the Mean Squared Error (MSE) over the masked targets predictions. Throughout training, we vary the weight used to balance out these two losses, starting with equal weights at the beginning of training, and annealing progressively the token prediction objective with a cosine schedule over the course of training.

We train our model with the AdamW optimizer [Loshchilov and Hutter, 2019]. We experimented with various early stopping criteria to train our model by setting aside a validation set, monitoring the validation MSE or Spearman's rank correlation between prediction and targets, varying the patience parameter (e.g., 3, 5 and 10) but obtained systematically better performance on the downstream tasks from our 'validation benchmark' when instead training for a fixed number of training steps (10k gradient steps). All training hyperparameters are summarized in Table 5.

Lastly, before training a ProteinNPT model on a given assay, we compute and persist to disk the sequence embeddings for all mutated proteins in that assay. During training, we load from disk the embeddings corresponding to each mini batch. Since the parameters of the model used to extract sequence embeddings are frozen, this procedure is strictly equivalent to computing embeddings on the fly. However, training is much faster as embeddings are computed only once and loading from disk is relatively fast. Additionally, we can work with larger training batches as memory requirements are lower – computing embeddings with large protein language models can necessitate substantial amounts of GPU memory.

## B.4 Inference

During the inference process, our objective is to make predictions for unlabeled protein sequences. These sequences are handled in our model using the same approach as we employed for sequences with masked targets during training. The input mini-batch is also comprised of labeled sequences from the training set, which are used to support the predictions of unlabeled instances. Ideally we would fit the entire training set when making predictions. However, this approach is often hindered by GPU memory bottlenecks in practice when the training set is sufficiently large. To that end, we set an upper bound $M$ on the maximum number of training sequences we can leverage at inference time: if the training dataset is smaller than $M$, we use all training sequences with no filtering; if the training dataset is larger, we sample at random $M$ sequences with no replacement. As seen in ablations above, we choose a value of $M = 1k$ in practice as larger values only yield marginal gains on downstream tasks performance. We also experimented with a sampling scheme with replacement in earlier ablations, but obtained systematically comparable performance with the scheme described above. The Regression Transformer [Born and Manica, 2023] can be seen as a special case of our architecture in which column-wise attention is ablated. In our experiments, this however results in a significant performance drop (Table 2).

## B.5 Difference with axial transformer, MSA Transformer and Non-parametric transformers

Our work builds on prior work from the self-attention literature, in particular the Axial Transformer [Ho et al., 2019], MSA Transformer [Rao et al., 2021] and Non-Parametric Transformer [Kossen et al., 2022].

Focusing on image and video modeling tasks, the Axial Transformer introduced the concept of axial attention, alternating in a given self-attention layer between row-wise attention and column-wise attention. We use this concept throughout our architecture to operate self-attention across residues (in embedding layers), across

residues and labels (in ProteinNPT layers), across sequences of a retrieved MSA (in embeddings layers) and across labeled observations (in ProteinNPT layers).

The MSA Transformer leveraged axial attention to learn protein sequence representations by training across a large number of Multiple Sequence Alignments (MSAs). Axial attention is used to alternatively learn dependencies across residues in the protein sequences, and across MSA sequences at a given position. The MSA Transformer is a fully unsupervised architecture and does not leverage available labels. Authors also show that a regression trained on attention maps of the MSA transformer achieve strong performance in predicting the residues that are in contact in the tertiary structure of the protein. It is an integral part of our ProteinNPT architecture, allowing us to obtain high-quality embeddings for each residue, which are then leveraged in the subsequent ProteinNPT layers.

Focusing on tabular datasets, Non-Parametric transformers (NPTs) leveraged axial attention to learn dependencies across features and labels of the tabular input, and across labeled instances. ProteinNPT differs from and extends NPTs in the following ways:

- In lieu of tabular inputs, we focus on a **different data modality** (protein sequences);
- Given the scarcity of labels involved with respect to the complexities of the underlying objects and their relations with the targets of interest, our architecture is **semi-supervised** and we leverage protein embeddings pre-trained on large quantities of unlabeled natural sequences;
- In our final architecture, we operate **tri-axial** self-attention as described above (instead of bi-axial self-attention in MSA Transformer and in standard NPTs);
- We introduce the concept of **auxiliary labels** which is critical for us to achieve strong performance, but it also broadly applicable beyond the protein-specific use cases we focus on in this work (§ 3);
- As discussed in Appendix C.1, we leverage **tied row attention**;
- We explore the **multi-task learning** setting (§ 4.5 and Appendix F);
- We investigate **uncertainty quantification** to support the Bayesian optimization tasks (Appendix G.1).
- We demonstrate the ability of our model to support **conditional sampling** of new sequences – which is critically important to prioritize sensible points to subsequently score in silico (§ 5.2);

## C   Baselines details

### C.1   Model architecture

As discussed in § 4.2, our first baseline is the top performing model that was introduced in Hsu et al. [2022], which learns a ridge regression on top of One-Hot-Encoded (OHE) representations of input protein sequences and unsupervised fitness predictions from DeepSequence [Riesselman et al., 2018]. We then sought to improve upon this initial baseline in two ways: 1) superior zero-shot fitness predictors 2) improved protein representations.

For the former, we ran ablations in which we replaced the zero-shot fitness predictions from DeepSequence with more recent alternatives: ESM-1v [Meier et al., 2022], MSA Transformer [Rao et al., 2021], Tranception [Notin et al., 2022a] and TranceptEVE [Notin et al., 2022b](see Appendix C.2). We generally see that gains in zero-shot fitness predictions translate into increased performance of the corresponding semi-supervised 'OHE - augmented' models.

For the latter, we build on the extensive literature [Heinzinger et al., 2019, Alley et al., 2019, Dallago et al., 2021, 2022, Elnaggar et al., 2023b, Stärk et al., 2021] of semi-supervised models for protein property prediction. These works typically first extract embeddings for labeled protein sequences with a large pre-trained protein language model. Given the dimensionality of the resulting full sequence embeddings – relatively large in practice compared with the size of the training data, a dimensionality reduction technique is typically used. The most commonly used approach consists in applying a *mean pooling* [Alley et al., 2019] operator across all the tokens of the input sequences. After pooling, a non-linear transform (e.g., MLP, CNN) is used to learn complex dependencies between the corresponding protein representations and the targets of interest. Albeit simple, this approach has lead to strong performance across various benchmarks [Dallago et al., 2022]. We then propose to combine these ideas with the zero-shot fitness prediction augmentations from Hsu et al. [2022] to obtain superior baselines. The corresponding baselines are referred to as 'Embeddings - Augmented'.

We carry out extensive ablations to optimize this new baseline further, and present a summary of the main variants tested in the next subsection (Appendix C.2). Our final top-performing baseline ('Embeddings - Augmented (MSA Transformer)') is based on embeddings obtained with the MSA Transformer, fed into a CNN layer with kernel size 9, followed by a ReLU non-linear activation, mean-pooled across sequence length, concatenated with zero-shot fitness predictions, and then passed to a linear projection for the final prediction. Following Hsu et al. [2022], the L2-penalty on the zero-shot fitness prediction parameter is set to a much lower value ($10^{-8}$) compared with other parameters ($5.10^{-2}$). We summarize details of the final architecture in Table 9.

Table 9: **Embeddings Augmented - Architecture details**

| Hyperparameter | Value |
|---|---|
| Embedding dimension | 768 |
| CNN kernel size (post embedding) | 9 |
| Weight decay (fitness pred.) | $10^{-8}$ |
| Weight decay (others) | $5.10^{-2}$ |
| Dropout | 0.1 |
| Max context length | 1024 |

Table 10: **Embeddings Augmented - Training details**

| Hyperparameter | Value |
|---|---|
| Training steps | 10k |
| Learning rate warmup steps | 100 |
| Peak learning rate | $3.10^{-4}$ |
| Optimizer | AdamW |
| Gradient clipping norm | 1.0 |
| Learning rate schedule | Cosine |
| Training batch | 64 |

Table 11: **Baselines ablations - Augmented OHE** Spearman's rank correlation between model predictions and experimental measurements.

| CV scheme | No Aug. | DS | ESM1v | MSAT | Tranception | TranceptEVE |
|---|---|---|---|---|---|---|
| Random | **0.572** | 0.510 | 0.508 | 0.494 | 0.499 | 0.521 |
| Modulo | 0.033 | 0.403 | 0.405 | 0.396 | 0.390 | **0.423** |
| Contiguous | 0.105 | 0.411 | 0.418 | 0.402 | 0.396 | **0.434** |
| Average | 0.237 | 0.442 | 0.444 | 0.431 | 0.428 | **0.459** |

As seen in Table 1, our improved baseline ('Embeddings - Augmented (MSA Transformer)') outperforms all OHE-based variants across all cross-validation schemes, including the ones augmented with the current best-in-class zero-shot fitness predictors. The lift is particularly important on the 'Contiguous' and 'Modulo' cross-validation schemes, for which the non-augmented OHE baseline performs just as poorly as random guessing, and the augmented equivalents perform on par with zero-shot fitness predictors.

## C.2 Ablations

We present in this section the main ablations conducted to give rise to our various baselines. We use the same subset of 8 DMS assays as described in Appendix B.2.

The first analysis (Table 11) looks at the effect of the model used for the zero-shot fitness predictions in OHE baselines. We find that the better the underlying zero-shot fitness predictor, the higher the performance of the corresponding OHE augmented model.

The second analysis (Table 12) investigates various pre-trained protein language model, and concludes that representations obtained with the MSA Transformer are the most robust across the various cross-validation schemes.

The third analysis (Table 13) compares several of the non-linear transforms that have been suggested by the prior literature to predict protein properties based on the sequence embeddings. More specifically, we investigated L2-penalized regression [Dallago et al., 2022], shallow Multi-Layer Perceptron (MLP) [Elnaggar et al., 2020], Convolutional Neural Network (CNN) [Dallago et al., 2022], ConvBERT [Elnaggar et al., 2023b] and Light Attention [Stärk et al., 2021]. We obtain our best results with a CNN layer with a kernel of size 9.

Lastly, we investigate the effect of the size of the training data on the model performance to assess the ability of the ProteinNPT architecture to generalize to shallow training settings (Table 8). We do not observe a drop in performance on datasets with fewer labeled instances relative to assays with more labels available during training, confirming the ability of the architecture to mitigate overfitting risks via its hybrid learning objective.

Table 12: **Baselines ablations - Protein sequence embeddings** Spearman's rank correlation between model predictions and experimental measurements.

| CV scheme | Tranception | ESM1v | MSAT |
|---|---|---|---|
| Random | **0.646** | 0.633 | 0.624 |
| Modulo | 0.498 | 0.534 | **0.553** |
| Contiguous | 0.413 | 0.487 | **0.509** |
| Average | 0.519 | 0.551 | **0.562** |

Table 13: **Baselines ablations - Non-linear transform post embedding** Spearman's rank correlation between model predictions and experimental measurements. We perform this ablation on the Random CV scheme only.

| Transform | Spearman |
|---|---|
| Light attention | 0.405 |
| Linear | 0.480 |
| MLP | 0.509 |
| ConvBERT | 0.617 |
| CNN | **0.624** |

## C.3 Model training

For all baselines ('OHE' and 'Embeddings'), the loss we optimize is the MSE loss between property predictions and true labels. Similarly to what we discussed in Appendix B.3, we train our various baselines for 10k steps, using the AdamW optimizer with a cosine learning rate schedule. All training hyperparameters are summarized in Table 10. At train time we find it helpful to pre-compute the MSA Transformer embeddings of all labeled examples to prevent GPU memory bottlenecks and speed up training. Following Rao et al. [2021], the embeddings for each DMS sequence are computed by first filtering sequences in the corresponding MSA with HHFilter [Steinegger et al., 2019] to ensure minimum coverage of 75% and maximum sequence identity of 90%, and then performing a weighted sampling of the MSA to extract 384 natural sequences. The sampling weights are based on the procedure described in Hopf et al. [2017b].

# D    Property prediction experiments for single mutants

## D.1    Experiment details

In this analysis we seek to assess the ability of various models to accurately predict the properties of single amino acid substitutions. We performed this analysis on all single mutants from the 100 assays from the extended ProteinGym substitution benchmark, and considered the 3 cross-validation schemes described in Appendix A.2.

We compared the performance of the following models:

- **Zero-shot (MSA Transformer)**: zero-shot predictions obtained with the MSA Transformer (ensembling over 5 weighted samples from the corresponding MSA, as described in Meier et al. [2022]);

- **OHE**: ridge regression on trained one-hot-encodings of the input protein sequences;

- **OHE - Augmented (DeepSequence)**: ridge regression on trained one-hot-encodings of the input protein sequences and zero-shot predictions from DeepSequence [Riesselman et al., 2018];

- **OHE - Augmented (MSA Transformer)**: ridge regression on trained one-hot-encodings of the input protein sequences and zero-shot predictions from the MSA Transformer [Rao et al., 2021];

- **Embeddings - Augmented (MSA Transformer)**: ridge regression on the concatenation of mean-pooled embeddings from the MSA Transformer and zero-shot predictions from the MSA Transformer (as described in Appendix C);

- **ProteinNPT**: final ProteinNPT architecture as described in § 3.

For brevity, we use the following shorthands in the Figures and Tables throughout this manuscript:

- **DS** stands for DeepSequence
- **MSAT** stands for MSA Transformer
- **PNPT** stands for ProteinNPT
- **Embed.** stands for Embeddings
- **Aug.** stands for Augmented

## D.2    Detailed performance results

We report the DMS-level and average performance for each cross-validation scheme in Fig 6 (Random cross-validation scheme), Fig. 7 (Modulo cross-validation scheme) and Fig. 8 (Contiguous cross-validation scheme). We also report the standard error for the Spearman's rank performance metric in Table 14, which we compute via non-parametric bootstrap of the difference between the Spearman performance of a given model and that of

Table 14: **Fitness prediction performance**. We compute the Spearman's rank correlation between model predictions and experimental measurements, averaged across the 100 DMS assays from the extended ProteinGym benchmark. DS and MSAT are shorthand for DeepSequence and MSA Transformer respectively. The Spearman for the MSA Transformer in the zero-shot setting is 0.41 (independent of CV split).

| Model name | Contiguous | Modulo | Random | Average |
|---|---|---|---|---|
| OHE | 0.08 (0.02) | 0.02 (0.02) | 0.54 (0.01) | 0.21 (0.02) |
| OHE - Augmented (DS) | 0.40 (0.01) | 0.39 (0.01) | 0.48 (0.01) | 0.42 (0.01) |
| OHE - Augmented (MSAT) | 0.41 (0.01) | 0.40 (0.01) | 0.50 (0.01) | 0.44 (0.01) |
| Embeddings - Augmented (MSAT) | 0.47 (0.01) | 0.49 (0.01) | 0.57 (0.01) | 0.51 (0.01) |
| ProteinNPT | **0.48 (0.00)** | **0.51 (0.00)** | **0.65 (0.00)** | **0.54 (0.00)** |

Table 15: **Multiple mutants fitness prediction - Performance summary.** We report the aggregated performance of ProteinNPT and other baselines listed in Appendix D.1, measured by the Spearman's rank correlation $\rho$ and Mean Squared Error (MSE) between model scores and experimental measurements. The standard error of the average performance is reported in parentheses.

| Model name | Spearman ↑ | MSE ↓ |
|---|---|---|
| Zero-shot (MSAT) | 0.367 (0.002) | – |
| OHE | 0.717 (0.002) | 0.541 (0.008) |
| OHE - Augmented (DeepSequence) | 0.719 (0.001) | 0.658 (0.106) |
| OHE - Augmented (MSAT) | 0.722 (0.001) | 0.516 (0.008) |
| Embeddings - Augmented (MSAT) | 0.695 (0.002) | 0.479 (0.008) |
| ProteinNPT | **0.788 (0.002)** | **0.259 (0.001)** |

the best overall model (i.e., ProteinNPT), computed over 10k bootstrap samples from the set of proteins in the extended ProteinGym benchmark.

We observe that ProteinNPT outperforms all baselines across the different cross-validation schemes. The largest performance lift is observed with the Random cross-validation as the model fully benefits from its column attention modules by learning from mutants that occurred at the same positions in the training set.

# E    Property prediction experiments for multiple mutants

## E.1    Experiment details

In this analysis we investigate the ability of the different models listed in Appendix D.1 to predict the effects of multiple mutations. We carry out this analysis on the subset of 17 DMS assays with multiple mutants from the extended ProteinGym substitution benchmark. As discussed in § 4.4, since multiple mutants occur across several positions simultaneously, we only conduct this analysis based on the Random cross-validation scheme.

## E.2    Detailed performance results

We report aggregated performance and corresponding standard error for all models in Table 15. Performance is reported in terms of Spearman's rank correlation and Mean Squared Error (MSE) between model predictions and experimental measurements. In the zero-shot setting, a fitness predictor is only able to rank order mutants based on relative fitness, but is unable to predict the actual value of the experimental phenotype. Thus, we do not report MSE performance in the zero-shot setting and only provide Spearman performance. We also report DMS-level performance (Spearman) in Fig. 2 (left).

We find that ProteinNPT markedly outperforms all other baselines both in terms of Spearman's rank correlation and Mean Squared Error. As shown in Fig. 2 (right), this is the case both for aggregate performance and across all mutation depths.

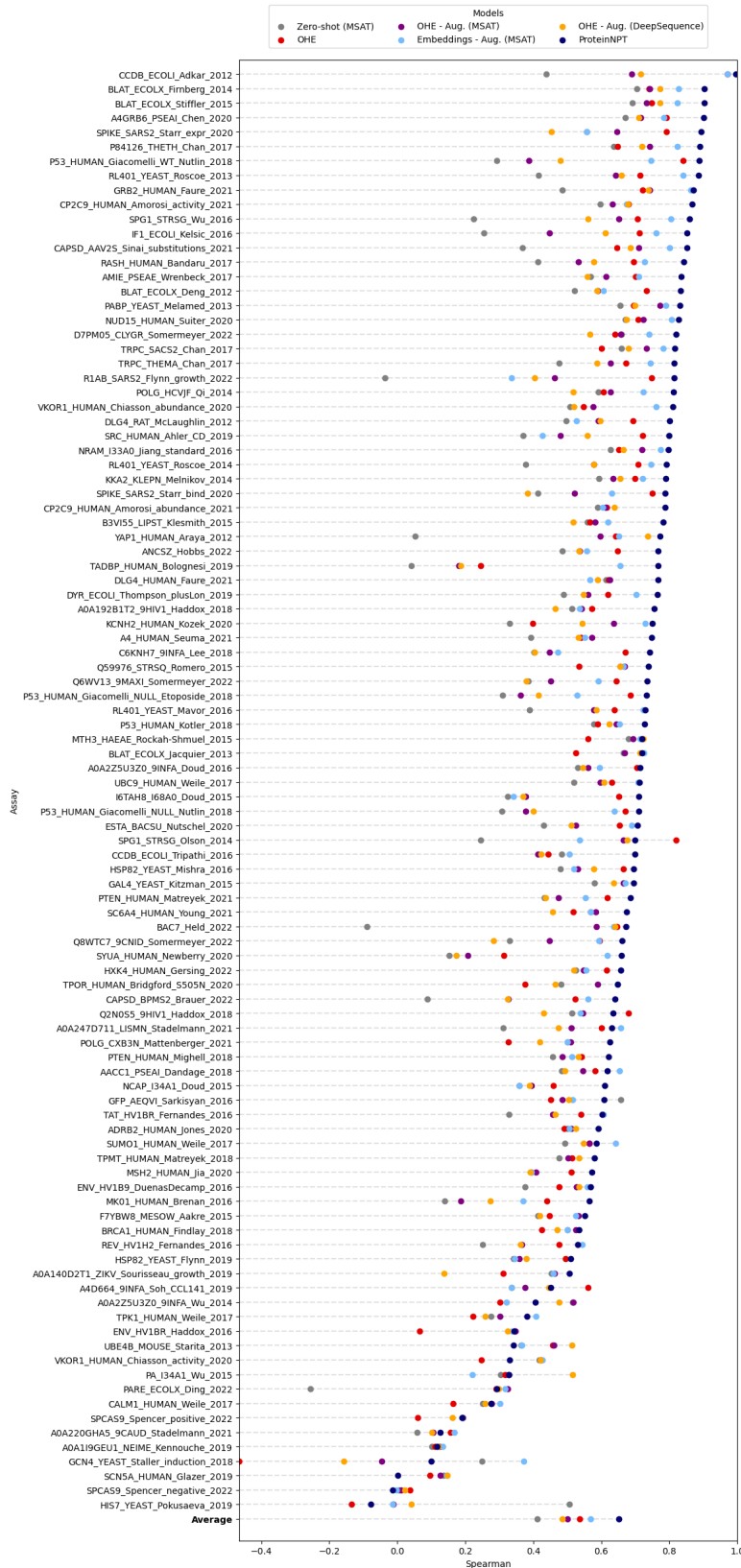

Figure 6: **Single mutants fitness prediction - Random cross-validation scheme** We report the DMS-level performance (measured by the Spearman's rank correlation $\rho$ between model scores and experimental measurements) of ProteinNPT and other baselines listed in Appendix D.1.

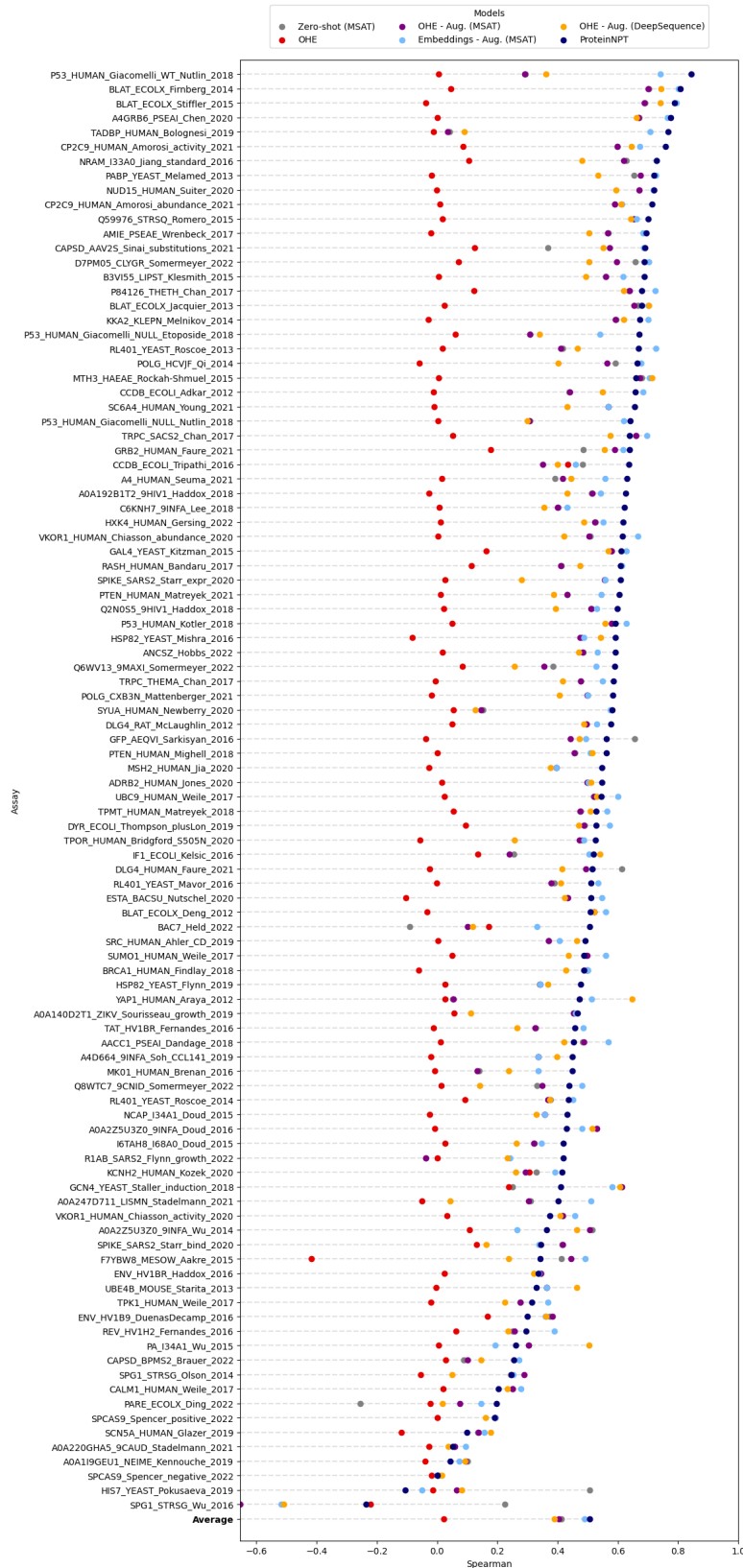

Figure 7: **Single mutants fitness prediction - Modulo cross-validation scheme** We report the DMS-level performance (measured by the Spearman's rank correlation $\rho$ between model scores and experimental measurements) of ProteinNPT and other baselines listed in Appendix D.1.

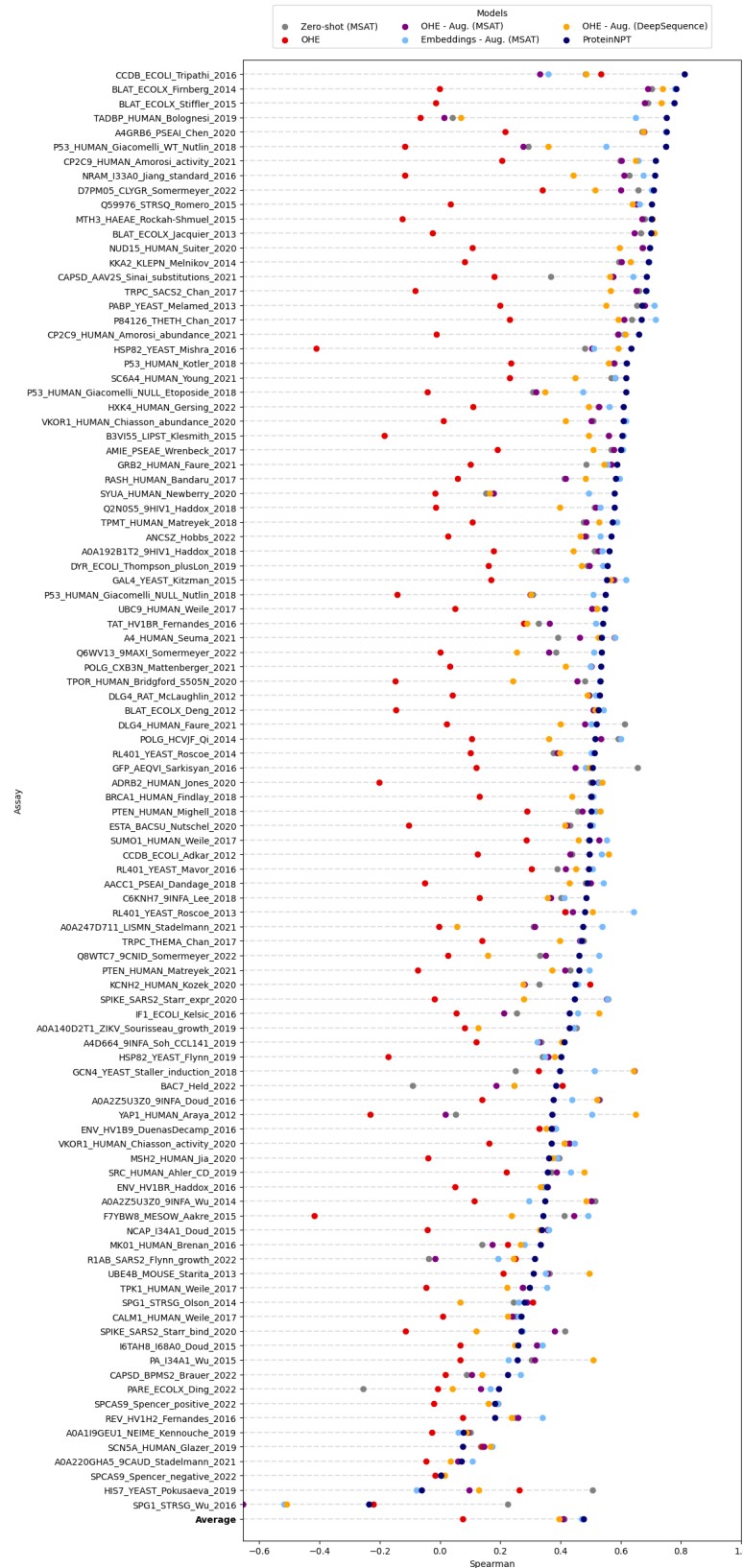

Figure 8: **Single mutants fitness prediction - Contiguous cross-validation scheme** We report the DMS-level performance (measured by the Spearman's rank correlation $\rho$ between model scores and experimental measurements) of ProteinNPT and other baselines listed in Appendix D.1.

Table 16: **Multiple properties prediction - Performance summary** We report the aggregated performance of ProteinNPT and other baselines listed in Appendix D.1, measured by the Spearman's rank correlation $\rho$ and Mean Squared Error (MSE) between model scores and experimental measurements.

| Model name | Spearman ↑ | | MSE ↓ | |
|---|---|---|---|---|
| | Assay 1 | Assay 2 | Assay 1 | Assay 2 |
| OHE | 0.609 | 0.601 | 0.917 | 0.917 |
| OHE - Aug. (DS) | 0.541 | 0.553 | 0.678 | 0.702 |
| OHE - Aug. (MSAT) | 0.524 | 0.533 | 0.687 | 0.718 |
| Embeddings - Aug. (MSAT) | 0.648 | 0.668 | 0.490 | 0.516 |
| ProteinNPT | **0.743** | **0.746** | **0.337** | **0.411** |

# F   Property prediction experiments for multiple properties

## F.1   Experiment details

We seek to evaluate the ability of the models to predict several properties simultaneously for the same mutated sequences. This is often highly important in practice as the majority of real-life protein design use cases call for the joint optimization of several properties, or the optimization of one property under constraints over other properties.

To that end, we extract from the extended ProteinGym substitution benchmark a subset of proteins for which several phenotypes have been measured. The seven proteins with at least two distinct measurements we considered were as follows:

- RL401 YEAST
- CCDB ECOLI
- VKOR1 HUMAN
- BLAT ECOLX
- P53 HUMAN
- PTEN HUMAN
- CP2C9 HUMAN

The three proteins with at least three measurements were as follows:

- RL401 YEAST
- BLAT ECOLX
- P53 HUMAN

We focus on the Random cross-validation scheme, and consider all possible mutants across the different assays for the same protein, which sometimes result in certain target values being missing for a subset of mutants.

For the various baselines we compare against, a separate linear head is used to predict each target based on the concatenation of the mean-pooled embeddings and zero-shot predictions. The loss that we optimize is the sum of the MSE losses for each target. When a subset of target values are missing for a given observation, we ignore the corresponding missing values in the loss computation and only leverage the loss for non-missing target values for these training instances.

In ProteinNPT, we simply add as input as many target columns as required, using the same masking procedure as described in Appendix B.3. Each target is then predicted by inputting the corresponding last-layer target embedding into a target-specific linear projection. The loss we optimize is the same composite loss which combines cross-entropy loss for the masked token prediction and the sum of MSE losses for each target prediction. When a subset of target values are missing for a given observation, we simply mask the corresponding input values. This leads to an implicit imputation scheme internally which contributes to further boosting the predictive performance.

## F.2   Detailed performance results

We report the performance of ProteinNPT and other baselines for the simultaneous predictions of two protein properties in Table 16, and three properties in Table 17. Consistent with the results from the above analyses, ProteinNPT outperforms all baselines across the different settings, both in terms of Spearman's rank correlation and MSE.

Table 17: **Multiple properties prediction - Performance summary.** We report the aggregated performance of ProteinNPT and other baselines listed in Appendix D.1, measured by the Spearman's rank correlation $\rho$ and Mean Squared Error (MSE) between model scores and experimental measurements.

| Model name | Spearman ↑ | | | MSE ↓ | | |
|---|---|---|---|---|---|---|
| | Assay 1 | Assay 2 | Assay 3 | Assay 1 | Assay 2 | Assay 3 |
| OHE | 0.694 | 0.600 | 0.720 | 2.718 | 2.792 | 2.721 |
| OHE - Aug. (DS) | 0.564 | 0.566 | 0.584 | 2.064 | 2.156 | 2.088 |
| OHE - Aug. (MSAT) | 0.534 | 0.538 | 0.539 | 2.087 | 2.172 | 2.147 |
| Embeddings - Aug. (MSAT) | 0.707 | 0.720 | 0.762 | 1.361 | 1.409 | 1.368 |
| ProteinNPT | **0.786** | **0.759** | **0.821** | **0.988** | **1.150** | **0.998** |

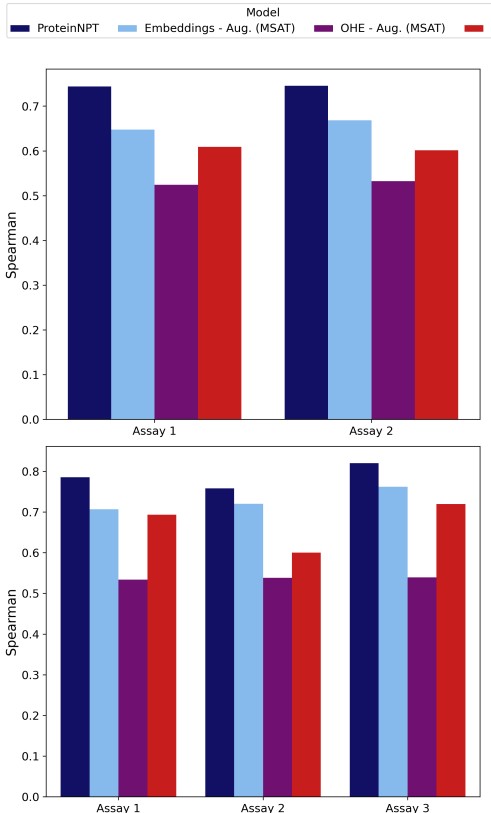

Figure 9: **Multiple target prediction** Avg. Spearman's rank correlation between model predictions and experimental measurements, for proteins with 2 or 3 distinct experimental measurements predicted simultaneously (respectively top and bottom plots).

## G    Bayesian optimization experiments

### G.1    Uncertainty quantification

We developed and compared three uncertainty quantification schemes:

1. **MC dropout:** We use a fixed inference batch at inference, apply Monte Carlo dropout Gal and Ghahramani [2015] and use the standard deviation of the prediction across multiple forward passes as our uncertainty metric;

2. **Batch resampling:** For a given point that we want to quantify the uncertainty of, we perform several forward passes by completing the input batches with a different sample (with replacement) of training

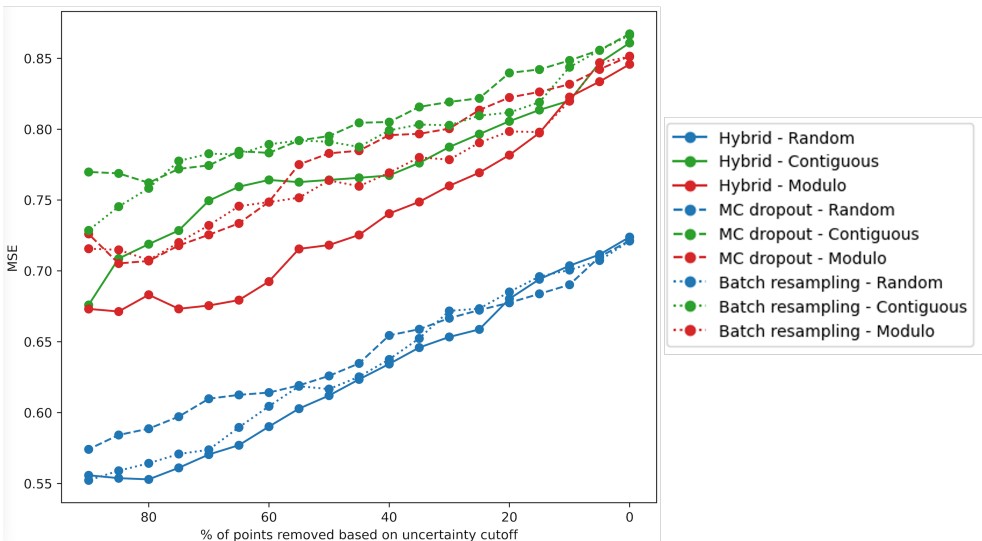

Figure 10: **Uncertainty calibration curves for each cross-validation scheme.** Uncertainty calibration curves plot a performance metric of interest, here MSE (y-axis), as a function of the proportion of points set aside based on their uncertainty (x-axis) (from right to left, we set aside an increasing fraction of the most uncertain points). To be properly calibrated, an uncertainty quantification metric should monotonically improve as we set aside an increasing proportion of the most uncertain points. We experiment with three different uncertainty quantification schemes: MC dropout, Batch resampling, and a hybrid scheme. For a fixed compute budget, the hybrid scheme delivers optimal performance across our three cross-validation schemes.

        points for each pass. No dropout is applied and we use the standard deviation across forward passes as our uncertainty metric;

3. **Hybrid scheme:** We combine the first two schemes together (ie, turning on dropout in the batch resampling sampling).

The hybrid scheme delivers superior performance in calibration curve analyses (Fig 10), for a fixed computational budget.

Uncertainty for all baseline models is calculated using MC dropout based on 25 forward passes through the corresponding models. The dropout rate is the same as in training, and is set to 0.1. Uncertainty for ProteinNPT models is obtained with the hybrid scheme based on 5 MC dropout samples for 5 distinct resampled batches, with both an attention dropout and an activation dropout set to 0.1. For each batch of sequences, we draw at random a set number (set to $M = 1k$, or to the total number of training points if less than that for Bayesian optimization experiments) of datapoints from the training set, which are required for non-parametric inference.

### G.2 Experiment details

We conduct the iterative protein redesign experiments on all DMS assays in the ProteinGym extended benchmark.

In this analysis we only consider single mutants, and exclude the multiple mutants that may be present in the original assay. We split assays in three groups based on the total number of datapoints they include:

- For assays with less than 1250 points, we start with 50 labels and acquire $\mathcal{N} = 50$ new labels at each iteration;
- For assays with more than 1250 points and less than 2500, we start with 100 labels and acquire $\mathcal{N} = 100$ new labels at each iteration;
- For assays with more than 2500 points, we start with 50 labels and acquire $\mathcal{N} = 200$ new labels at each iteration.

Note that at iteration 0, we therefore start with a few labeled examples, which explains why performance does not start at 0 on the performance plots. At each round, the available pool is constituted of all sequences minus the ones already selected. As such, we do not disregard any sequence, and all available sequences are scored using the upper confidence bound acquisition function. For robustness, we run each experiments three times, with

a different starting pool. We set the $\alpha$ parameter, controlling the exploration trade-off to 0.1, after optimising performance on a subset of 3 assays.

### G.3 Detailed performance results

We plot the recall rate of high fitness points (top 3 deciles) as a function of the number of batch acquisition cycles for all DMS assays in the extended benchmark. We find that ProteinNPT performs better than baselines on almost all assays. Moreover, when ProteinNPT does not perform well, it is usually because the assay is 'hard' to learn, because the property is intrinsically difficult to predict or experimental measurements are noisy (or both) – on these assays the performance of all baselines is close to that of a random acquisition strategy. This is for example the case for TAT HV1BR, CALM1 HUMAN and REV HV1H2 experiments. Lastly, we observe that ProteinNPT performs better for assays with a larger number of mutants. Since real-world design scenarios are less restricted than the in silico settings we consider here, it is therefore encouraging to see that ProteinNPT performs better when larger mutant pools are available.

## H Codebase and License

The entire codebase to train and evaluate ProteinNPT and the various baselines considered in this work is available on our GitHub repository (`https://github.com/OATML-Markslab/ProteinNPT`) under the MIT license.

## I Compute

All experiments carried out in this work were conducted in Pytorch, on A100 GPUs with either 40GB or 80GB of GPU memory.

## J Limits

As a semi-supervised pseudo-generative model, ProteinNPT is dependent on the number and quality of available labels used for training. The diversity of the 100 DMS assays from the extended ProteinGym benchmark provide a sense for the sensitivity of performance to various characteristics of the underlying protein family and available labels in the corresponding assay. While compute cost is typically a secondary consideration in protein design endeavours as it is dwarfed by the cost of wet lab experiments, one should note that ProteinNPT is more computationally expensive than the other baselines discussed in this work. This overhead directly depends on the protein sequence length and the number of training instances per mini-batch since both impact the compute and memory complexities of the row and column attention in ProteinNPT layers.

## K Ethics

The models introduced in this work can be used to support the design of novel proteins of any kind. As such we acknowledge the dual potential for both positive and negative applications. We commit to exercising caution in their deployment (we will not be releasing model parameters for instance) and we are dedicated to actively promoting their responsible use.

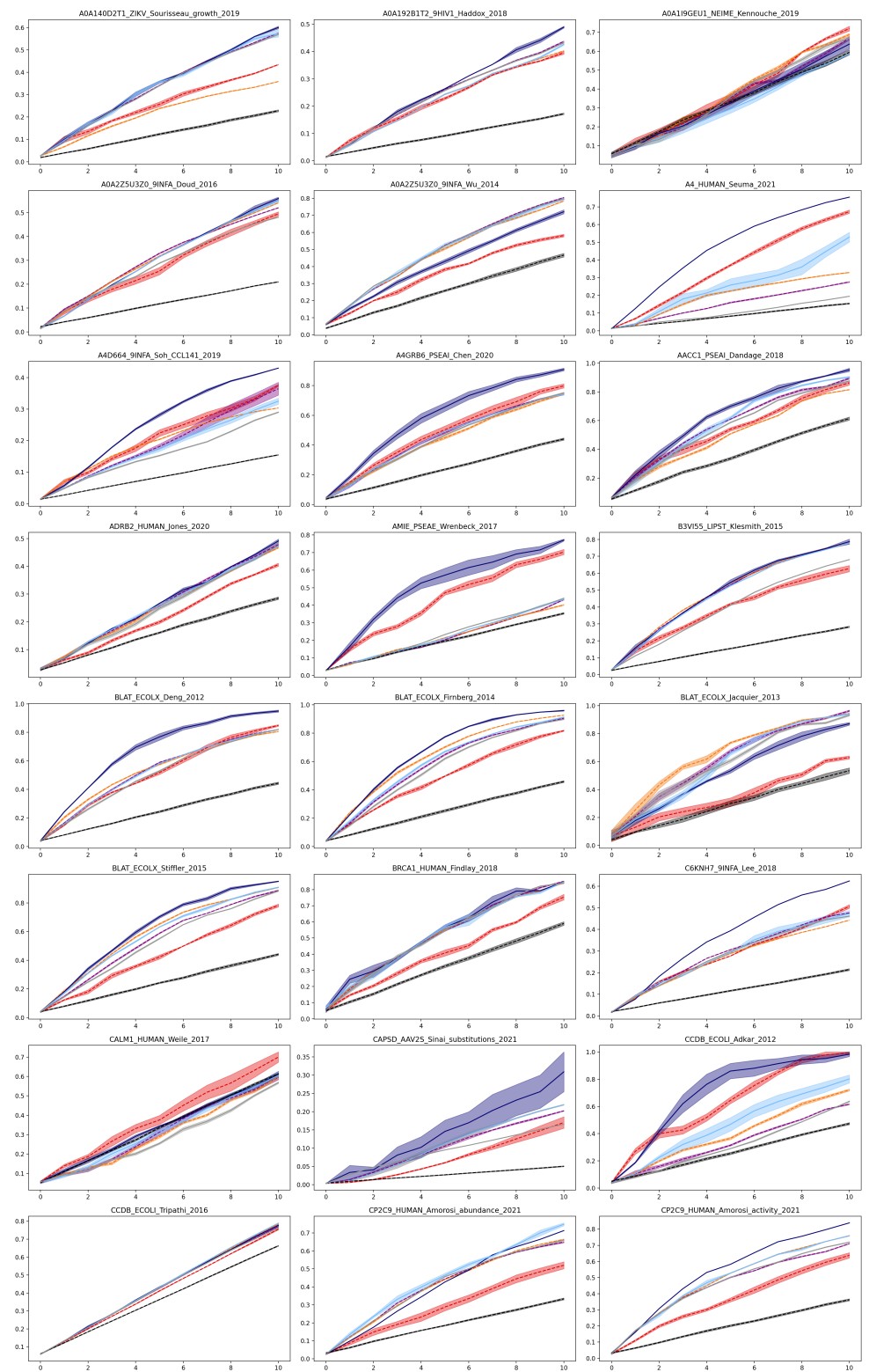

Figure 11: **DMS-level performance for iterative protein redesign experiments (Assays 1-24).**
We plot the recall rate of high fitness points (top 3 deciles) as the function of the number of batch acquisition cycles. The shaded regions represent the standard errors of each method.

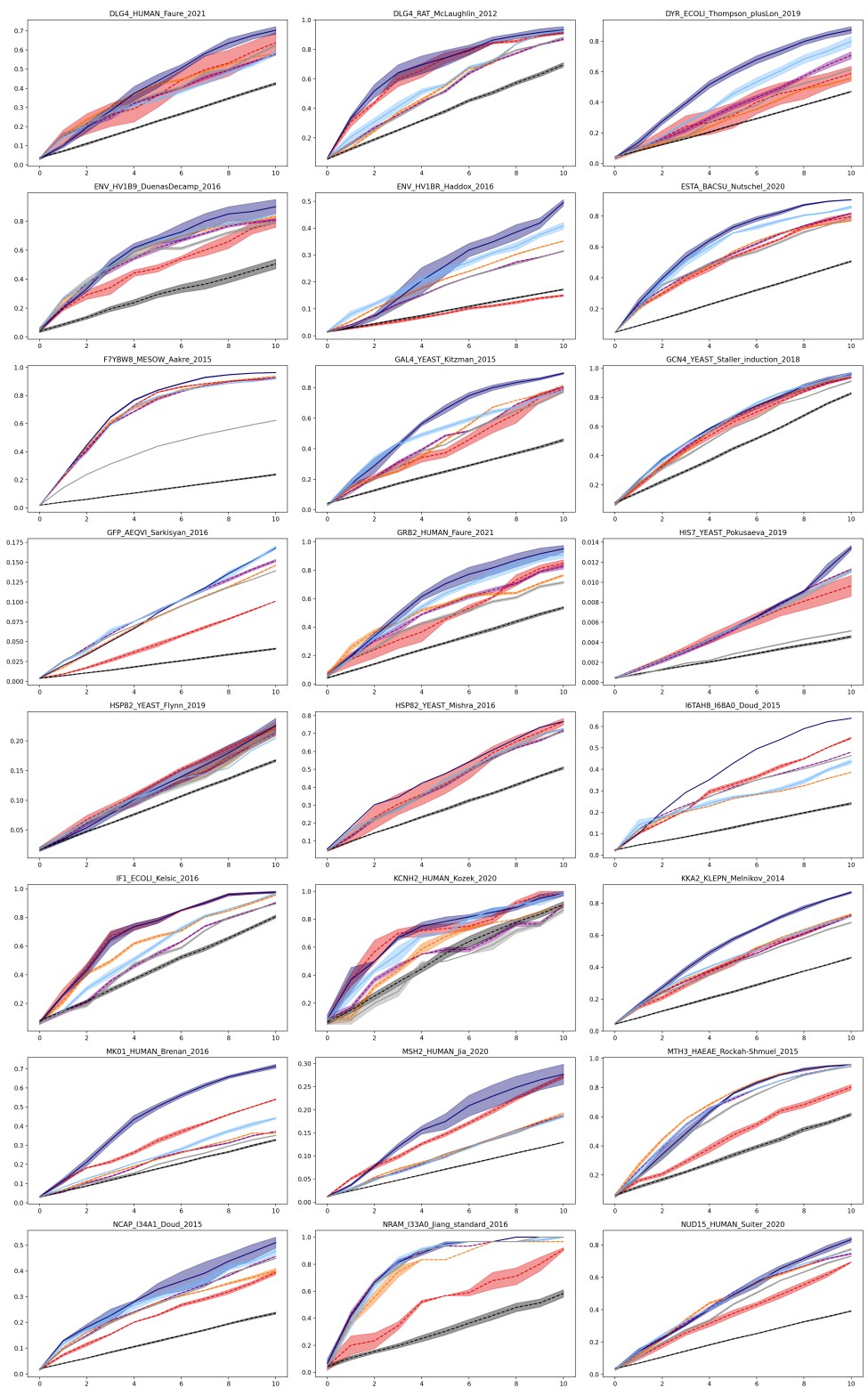

Figure 12: **DMS-level performance for iterative protein redesign experiments (Assays 25-48).** We plot the recall rate of high fitness points (top 3 deciles) as the function of the number of batch acquisition cycles. The shaded regions represent the standard errors of each method.

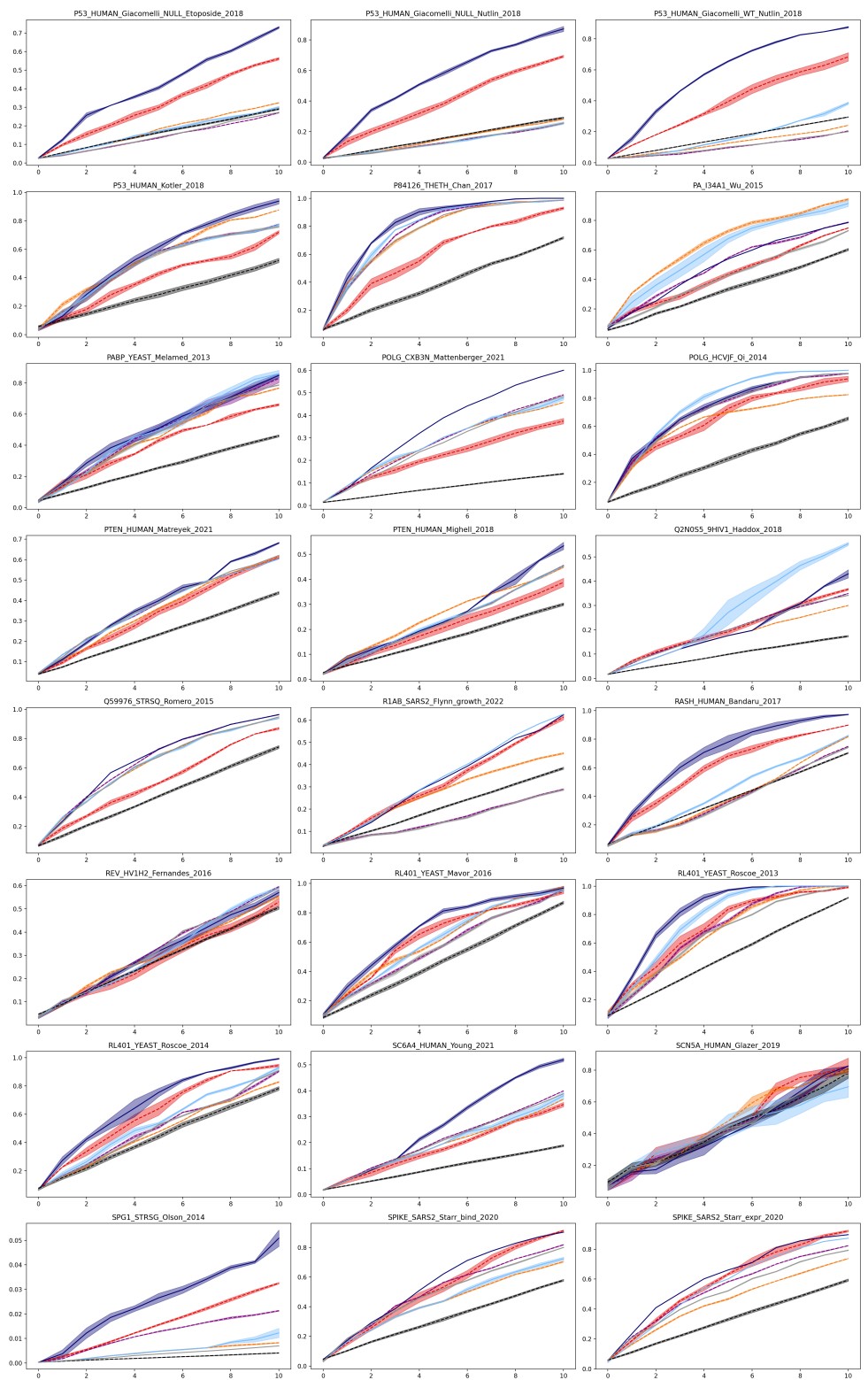

Figure 13: **DMS-level performance for iterative protein redesign experiments (Assays 49-72).**
We plot the recall rate of high fitness points (top 3 deciles) as the function of the number of batch
acquisition cycles. The shaded regions represent the standard errors of each method.

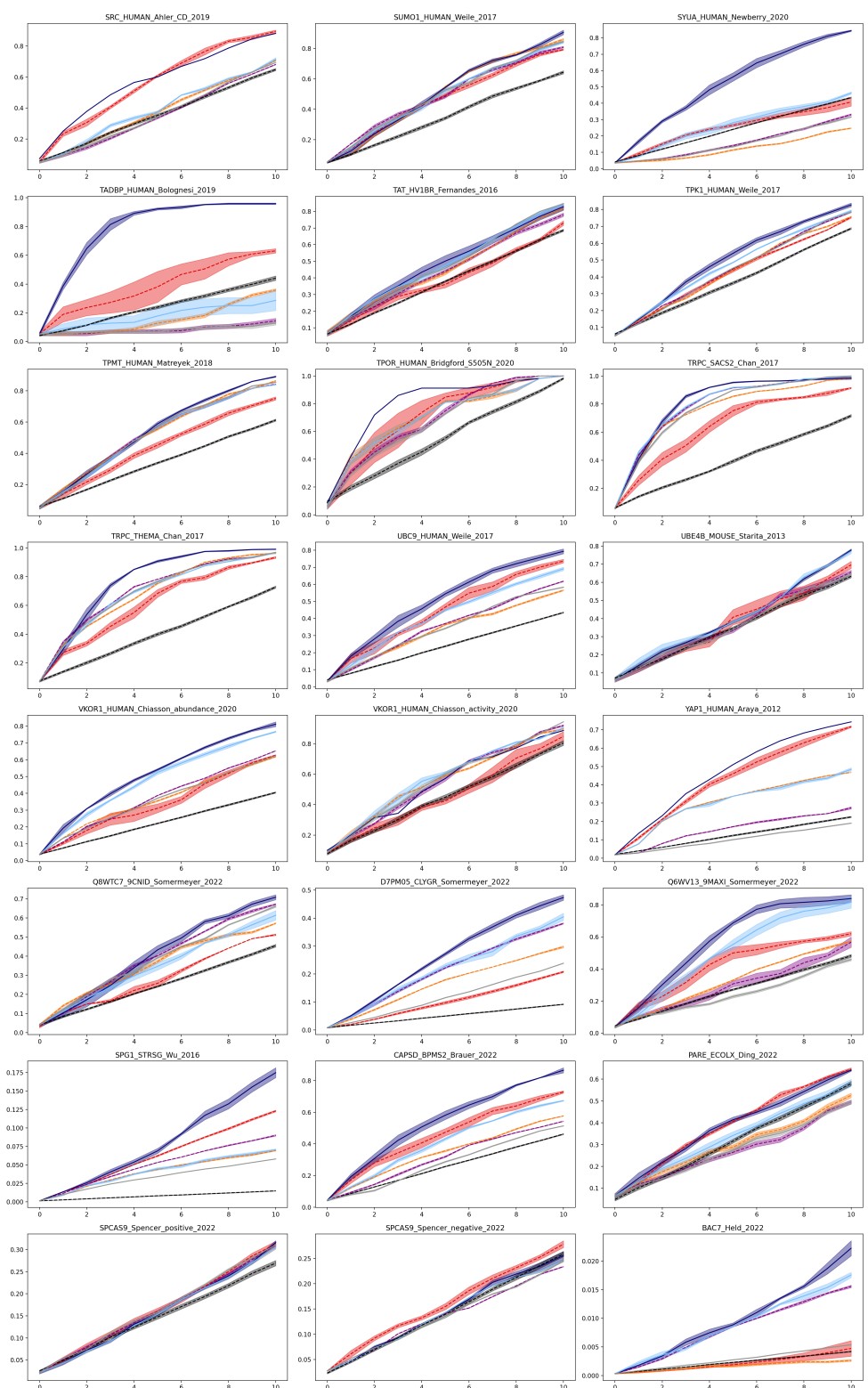

Figure 14: **DMS-level performance for iterative protein redesign experiments (Assays 73-96).**
We plot the recall rate of high fitness points (top 3 deciles) as the function of the number of batch acquisition cycles. The shaded regions represent the standard errors of each method.

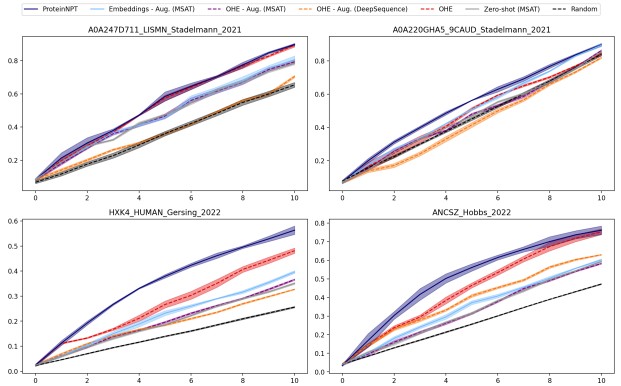

Figure 15: **DMS-level performance for iterative protein redesign experiments (Assays 97-100).** We plot the recall rate of high fitness points (top 3 deciles) as the function of the number of batch acquisition cycles. The shaded regions represent the standard errors of each method.

