# OpenReview forum: "ProteinNPT: Improving Protein Property Prediction and Design with Non-Parametric Transformers"
_NeurIPS.cc/2023/Conference — NeurIPS 2023 poster_

### Official Review · Reviewer_42SC · 2023-07-05

**Soundness:** 3 good
**Presentation:** 3 good
**Contribution:** 3 good
**Rating:** 6
**Confidence:** 3

**Summary:**

This paper introduced ProteinNPT, which is a transformer based semi-supervised learning model. By combining MSA transformer and Non-parametric transformer(NPT), it outperformed their baselines on property and mutation effect predictions. Then they tested their model on the protein design task and showed its potential to help on designing novel protein sequences with desired labels. In addition, its contributions also include the new extended benchmark dataset, ProteinGym, as part of their test set to evaluate their model. This paper leveraged large quantities of unlabelled natural sequences to pretrain the model to have a more informative embedding. Moreover, the tri-axial attention helps the model to learn relationships not only within the row, but also across the data points in a batch. All these build up together to boost the performance of the model on multiple downstream tasks.


**Strengths:**

The paper addresses an important topic which is the subject of much research. The flow of this paper is clear. It introduces the motivation first and then its solution to the problem fits the motivation nicely. In addition, the experimental design supports their claims by showing their model outperforming the baseline models in different downstrain tasks. Furthermore, they summarize their contributions and novelties clearly
The authors nicely demonstrate improvements in the quality of representation learning by comparing ProteinNPT with baselines methods on property prediction and mutation effect prediction tasks.
To enhance the claim of advantages of tri-axial attention and auxiliary labels, the paper further shows that it can succeed in the protein fitness prediction problem. In this task, the authors provide some examples to illustrate their good performance while their evaluation metrics show that their model outperforms the other models in general.
Using ablation studies, the authors demonstrate that their embedding can help in different cross validation schemes compared to other methods.
The paper well built an experiment with ProteinNPT + Bayesian optimization methods and showed its ability to redesign protein with desired properties in an iterative manner.


**Weaknesses:**

The paper did not illustrate the model design very clearly. The supplied figure is just too simplistic. The paper’s annotations and formulas give the audience a basic understanding of what the authors aim to do. However, it is hard to follow how the MSA transformer is introduced to the model. They should have more explicit illustration about how it is trained(i.e with what data) and where it is used as part of the ProteinNPT. In addition to that, it is vague about the training data for the MSA transformer in the baselines. It will be helpful for the audience to understand the experiment design if the paper can provide more details about how MSA sequences are generated(if used) and if the ProteinNPT used the same MSA transformer as the baseline.

The authors mention their novelties in different aspects. In fact, the dedicate a whole subsection for it, which helps detail the novelty. However, beyond the additional data and testing that come with it, most of the novelties appear to be subtle changes or directly borrowed ideas from MSA transformer or NPT. For example, the authors claim one of their novelties is a semi-supervised architecture but it is actually mentioned in NPT paper.

In the Figure 5 and 6, their model actually has similar performance or even lose to the DeepSequence and MSAT models in a lot of datasets. However, there is no analysis of the reason.

Minor comments:
Line 152, 167, 268: There is no “Fig. 4”.
Table 6: the second “100” should be “1000”
Define "MSA Transformer" line 91,94 then switch to "MSAT" line 100
"labelled" line 294
"a a sets" line 94
"is that is" line 178
"faction of test" line 197
Description line 291-296 unclear

Reference in line 400 does not appear correctly


**Questions:**

see above.
Also: Line 180: The idea of using informative other labels as features make sense. In reality they use MSAT predictions which makes their model like an ensamble of sorts. What happens if they take these labels out? How is performance affected?



**Limitations:**

see above

---

> ### Author Rebuttal · Authors · 2023-08-10
>
> **C1. The authors mention their novelties in different aspects. In fact, the dedicate a whole subsection for it, which helps detail the novelty. However, beyond the additional data and testing that come with it, most of the novelties appear to be subtle changes or directly borrowed ideas from MSA transformer or NPT. For example, the authors claim one of their novelties is a semi-supervised architecture but it is actually mentioned in NPT paper.**
>
> We summarize the various architectural contributions we made in the first comment to all reviewers. The use of auxiliary labels (section 4.4 and response to your comment C4), multiple optimization with NPTs (section 5.4), conditional sampling (section 4.5 and Fig. A in rebuttal pdf), uncertainty quantification (section 6, appendix G1, and Fig C and D in rebuttal) in NPTs are all novel and non-trivial contributions that have not been discussed in any prior work, in particular the original NPT paper.
>
> As for the very last point made by the reviewer, we would like to clarify that NPTs are actually not semi-supervised but purely supervised. A naive application of standard NPT to protein fitness prediction for instance would actually do very poorly, as strong performance requires high quality embeddings typically obtained by training protein language models on substantially larger datasets. To support this claim we ran an additional experiment in which we train a standard NPT where we rely on learned embeddings as opposed to pretrained embeddings from large protein language models, and remove auxiliary labels. We observe a significant performance drop (Table 1).
>
> **C2. In the Figure 5 and 6, their model actually has similar performance or even lose to the DeepSequence and MSAT models in a lot of datasets. However, there is no analysis of the reason.**
>
> This is a very important point in practice that we will clarify further in the text: the relative ranking of fitness predictors exhibits a lot of assay-to-assay variation, as has been observed both in the zero-shot (Riesselman et al., Laine et al., Notin et al.) or supervised settings (Dallago et al., Hsu et al.). Consequently, robust conclusions about the relative benefits of various model architecture require benchmarking across a wide range of assays. As indicated by reviewer yxtV in comment C5, several papers often focus on single assays, such as GFP, with others going up to the ~40 assays from the DeepSequence benchmark (Risselman et al), such as (Hsu et al), which we compare against abundantly in our work (e.g.,  OHE - Aug. DS). To our knowledge, no other work has analyzed performance or fitness predictors on a set as broad as covered in this work (100 assays).
>
> **C3. Minor comments**
>
> Thank you very much for flagging these. We now have corrected all these points in the manuscript.
>
> **C4. Line 180: The idea of using informative other labels as features make sense. In reality they use MSAT predictions which makes their model like an ensemble of sorts. What happens if they take these labels out? How is performance affected?**
>
> We refer the reviewer to the ablation analysis reported in Table 4 in Appendix B.2 which investigates the critical role of auxiliary labels to performance. We have also included these results in Table A of the pdf.
>
> **References**
>
> - Riesselman et al. “Deep generative models of genetic variation capture the effects of mutations.” Nature Methods (2018)
> - Laine et al. “GEMME: A Simple and Fast Global Epistatic Model Predicting Mutational Effects.” Molecular Biology and Evolution (2019)
> - Notin et al. “Tranception: protein fitness prediction with autoregressive transformers and inference-time retrieval.” ICML (2022)
> - Dallago et al. “FLIP: Benchmark tasks in fitness landscape inference for proteins.” NeurIPS (2021)
> - Hsu et al. “Learning protein fitness models from evolutionary and assay-labeled data.” Nature Biotechnology (2022)

---

> > ### Comment · Reviewer_42SC · 2023-08-13
> > **Thank the authors for the clarifications**
> >
> > We thank the authors for the clarifications. We did not find any major flaws with the paper before and therefore retain our previous (positive) score.

---

### Official Review · Reviewer_af1N · 2023-07-06

**Soundness:** 2 fair
**Presentation:** 2 fair
**Contribution:** 3 good
**Rating:** 6
**Confidence:** 4

**Summary:**

The paper researches for the purpose of iterative protein design. It first collects 13 additional fitness metrics to the ProteinGym benchmark. Then, it proposes ProteinNPT, an MSA-Transformer-based model for fitness prediction, and achieved state-of-the-art on ProteinGym. Finally, it applies the model to iterative protein redesign via Bayesian Optimization.

**Strengths:**

The experimental results are promising.

**Weaknesses:**

1. Novelty: The way adapts Transformer to property prediction is straightforward. The novelty is limited.
2. Transferability: Additional training is required for different fitness settings. At the same time, the architecture seems to be difficult to even generalize to closely correlated fitness settings, for the labels are plugged into the input. The transferability is limited.
3. Real-world application: Labeled data is required for training, which would bring wet-lab costs. Additionally, the method is based on MSAs consisting of mutated sequences, which tend to be gathered at a certain point in the protein space rather than cover much space, so the method seems unsuitable to be applied in the discovery stage. As for the optimization stage, I doubt whether the precision of the proposed model satisfies the requirements from biology researchers. Overall, I think the space for real-world application is limited.
4. Other suggestions: Exchange "left" and "right" in the text description of Fig. 1, and declare “OHE” as One-Hot-Encoded in the main paper.


**Questions:**

N/A

---

> ### Author Rebuttal · Authors · 2023-08-10
>
> **C1. Novelty: The way adapts Transformer to property prediction is straightforward. The novelty is limited.**
>
> We would like to first clarify that we do not adapt the standard Transformer architecture to property prediction, but rather apply a non-trivial variant of axial transformer as described in Section 4. We also refer the reviewer to the response we provided to reviewer 8GnD (C2) for equations to clarify the significant differences with standard Transformer and the first comment to all reviewers regarding novelty and contributions.
>
> **C2. Transferability: Additional training is required for different fitness settings [...] difficult to even generalize to closely correlated fitness settings, for the labels are plugged into the input. The transferability is limited.**
>
> Training protein-specific models on use-case specific labels is a staple of modern protein engineering methods (see response to the next question for a more detailed perspective on this). There is no such thing as fitness in the abstract but rather fitness at a specific temperature,  pH,  in different cells or in this manufacturing process. Therefore, in protein engineering, there is a huge unmet need for semi-supervised models that take in different kinds of labels in the same overall architecture. This is the ultimate generalizability for design goals. To the reviewer’s point, it would be nice if there were a method and data out there that would nicely generalize to accurately predicting all of these various design objectives in a unified architecture. But the reality of the field is that no such data or approach currently exists and that all the practical ML-driven protein engineering efforts over the past decade have relied on training ML models on use-case and property-specific labels, often across iterative design cycles such as the setting we described in section 6 of our paper. This is particularly true for efforts that involve the simultaneous optimization of multiple properties that may be at odds with one another. For instance the development of protein-based therapeutics is frequently faced with the difficult task of optimizing existing proteins that simultaneously have maximal binding with a specific target, minimal binding with everything else (the so-called de-immunization objective), operate at a certain pH, maintain thermostability.
> Our approach only requires commodity hardware to train the corresponding models on new labels. Our experiments demonstrate that the same model architecture and training procedure (all final experiments in section 5 are conducted with the same hyperparameters across) is general and performs well across a very wide range of protein properties including binding, thermostability, enzymatic activity, etc.
> The average wet lab costs typically dwarf compute costs. Any performance lift can save huge costs in wet lab experiments, so the relatively small computational cost involved to fine tune models on labels as we suggest is desirable in practice.
>
> **C3. Real-world application: Labeled data is required for training, which would bring wet-lab costs.**
>
> There is more and more publicly available labeled data that can be leveraged for design;  see for instance the collection of data in Protein Gym (Notin et al. 2022), FLIP and papers (Riesselman et al. 2018; Hopf et al. 2017; Shin et al. 2021; Livesey and Marsh 2022) that have used many such datasets. To date,  these kinds of datasets have been successfully used by unsupervised  ML methods  for validation as far to phenotype as clinical outcome (Frazer et al. 2021) and moving forward to designing for specific fitness with respect to conditions for manufacturing, one can imagine the iterative design of experiments we address in this paper.
>  As for real world applications - this and other generative models have already shown promise for real-world applications. For instance, the original evolutionary coupling models trained on homologous sequences demonstrated the ability of existing data with generative models to predict 3D structure from sequences (Marks et al. 2012),  AlphaFold (Jumper et al. 2021) and 3D contacts from MSA Transformer paper (Rao et al. 2021). As for existing.open sources data with labels where the sequences are  distant from a wild type -  there are also increasing numbers  published examples, e.g. chorismate mutases (Russ et al. 2020), GFPs (Weinstein et al. 2023,Gonzalez Somermeyer et al. 2022), AAV capsids (Riley et al. 2021), plastic eating enzymes (Lu et al. 2022), the mega-data set on thermostability mentioned above(Tsuboyama et al. 2023) and many others. Despite amazing progress in the past couple years in de novo design (Watson et al. 2022), zero-shot methods are - to date - unable to generate proteins with specific functions with a high success rate. So the question is not whether labels are needed -- they always are, but rather how to best leverage them in iterative design cycles. The current approaches to do that are either lightweight supervised methods based on hand-crafted protein features or, more recently, supervised methods that use embeddings from large protein language models as input.
>
> **C4. Additionally, the method is based on MSAs consisting of mutated sequences [...] Overall, I think the space for real-world application is limited.**
>
> This is a misunderstanding: the MSAs that we use in this study are always composed of homologous natural sequences only, and never include mutated sequences. Furthermore, the MSA Transformer used to obtain sequence embeddings is trained across millions of protein families (MSAs), precisely as a means to generalize to unseen regions of sequence space. Lastly, the performance lift provided by supervision over zero-shot methods is substantial in practice (Fig D in pdf), so it is not clear how one could achieve superior performance while at the same time not using labels.
>
> **C5. Other suggestions**
>
> Thank you for flagging these. We fixed both in the revision.

---

> > ### Comment · Reviewer_af1N · 2023-08-11
> > **Thanks for the Rebuttal**
> >
> > I am updating the rating from 4 to 6.

---

### Official Review · Reviewer_8GnD · 2023-07-06

**Soundness:** 3 good
**Presentation:** 2 fair
**Contribution:** 3 good
**Rating:** 5
**Confidence:** 3

**Summary:**

his paper introduces a non-parametric transformer variant called ProteinNPT, which is utilized for protein property prediction and design tasks. The study compares ProteinNPT with re-implemented top-performing baselines on the extended ProteinGym benchmark.

The objective of the paper is to address challenges faced in protein engineering, including the vast design space, sparse functional regions, and limited availability of labels. The results demonstrate that the proposed ProteinNPT outperforms all the compared methods in various protein property prediction tasks and also shows potentials in the protein design task.

**Strengths:**

This paper introduces ProteinNPT, a non-parametric variant of transformers, which is employed for protein property prediction and design tasks. The authors extend the ProteinGym benchmark and re-implement several top-performing baselines for comparison with ProteinNPT. The primary focus of ProteinNPT is to tackle the challenge of limited availability of labels, and it can be effectively combined with state-of-the-art protein language models.

The problems investigated in this paper hold significant importance and appeal from a biological perspective. The authors have expanded the ProteinGym benchmark and conducted a comprehensive experimental analysis, with very detailed and sufficient results provided in both regular paper and Appendix. In conclusion, the proposed ProteinNPT offers substantial value and can contribute to the advancement of related research.

**Weaknesses:**

The paper's presentation style may pose challenges for readers in terms of comprehension. It is important to note that this comment does not reflect the quality of the writing itself, but rather the accessibility and reader-friendliness of the content. In order to fully comprehend the paper, readers are required to go through the entirety of the appendix. In summary, it is good for the paper to be as self-contained as possible. For instance, "the difference with axial transformer, MSA Transformer and Non-parametric transformers" would be better suited within the main body of the paper rather than relegating it to the Appendix. In general, it is crucial to explicitly state the differences between the proposed method and other comparable approaches when the leaving space of paper is enough.

Furthermore, it would be beneficial to include additional mathematical formulations pertaining to ProteinNPT in Section 4, rather than solely relying on the architecture diagram for explanation.

**Questions:**

I am interested in the part of model training (Section 4.2 and Appendix B.3). It would be beneficial if the authors could provide a detailed description of the multi-task optimization process. For instance, could you explain how the annealing process is implemented for the token prediction objective? Additionally, could you elaborate on the use of a cosine schedule throughout the training process?

Minor Miscellaneous Suggestions

- Line 152 and 268: 'Fig 2', not 'Fig 4'?
- Table-1: The full name of abbreviation "OHE" needs to be added.

**Limitations:**

No obvious limitations.

---

> ### Author Rebuttal · Authors · 2023-08-10
>
> **C1. The difference with axial transformer [...] better suited within the main body of the paper rather than relegating it to the Appendix.**
> We have integrated the points made in Appendix B6 within Sec. 2 as suggested.
>
> **C2. It would be beneficial to include additional mathematical formulations [...] architecture diagram for explanation.
> We thank the reviewer for the great suggestion, and are including the main mathematical equations for the ProteinNPT architecture below and in section 4  in the revised manuscript.**
>
> **ProteinNPT architecture**
>
> Let $(X^{\text{full}},Y^{\text{full}})$ be the full training dataset where $X^{\text{full}} \in \left[[ 1,20 \right]]^{N.L}$ are protein sequences (with N the total number of labeled protein sequences and L the sequence length), and $Y^{\text{full}} \in \mathbb{R}^{N.T}$ the corresponding property labels (where $T$ is the number of distinct such labels, including auxiliary labels). During training, for each gradient step, we sample at random a mini-batch $(X,Y)$ and pass both as \emph{input} to the ProteinNPT architecture. In accordance with our denoising modeling objective, we mask a fixed proportion of input tokens and labels (15\% for both). We then embed sequences and labels separately, with a pretrained and frozen protein language model and a learned label embedding respectively, each amino acid token and label being embedded in a vector of dimension $D$. We obtained our best results (Appendix B2, Table 10) embedding protein sequences with the MSA Transformer, which applies axial attention on a Multiple Sequence Alignment (MSA) for the corresponding family (attention across amino acid tokens and across natural sequences in the MSA). After concatenating the resulting token and label embeddings, we obtain a unique tensor $Z \in \mathbb{R}^{(N.(L+T).D)}$ that is then fed into several ProteinNPT layers.
> Each ProteinNPT layer applies successively \emph{row-attention}, \emph{column-attention} and a feedforward layer. Each of these transforms is preceded by a LayerNorm operator $LN(.)$ and we add residual connections to the output of each step. For the multi-head row-attention sub-layer, we linearly project embeddings for each labeled sequence $n \in \left[[ 1,N\right]] $ for each attention head $i \in \left[[1,H\right]]$ via the linear embeddings $Wr_{i}^{K}$, $Wr_{i}^{Q}$ and $Wr_{i}^{V}$ respectively. Mathematically, we thus have:
> $\text{Row-Att}(Z) = Z + \text{Tied-MHSA}(LN(Z)) = Z + \text{concat}(O_1,O_2,...,O_H).W^O \in \mathbb{R}^{N.L.D}$ where the concatenation is performed row-wise, $W_O$ mixes outputs from different heads, $O_i = \text{Tied-Att}(Z.Wr_{i}^{Q}, Z.Wr_{i}^{K}, Z.Wr_{i}^{V})$ and with the tied row-attention is as defined in Rao et al. $ \text{Tied-Att}((Q_n,K_n,V_n)) = \text{softmax}(\sum_{n=1}^{N}
>  ((Q_n.K_n^{T}) / \sqrt{N.D}).V_n)$.
>
> We then apply column-attention as follows:
> $\text{Col-Att}(Z) = Z + \text{MHSA}(LN(Z)) = Z + \text{concat}(P_1,P_2,...,P_H).W^P \in \mathbb{R}^{N.L.D}$ where the concatenation is performed column-wise; $W_P$ mixes outputs from different heads, $P_i = \text{Att}(Z.Wc_{i}^{Q}, Z.Wc_{i}^{K}, Z.Wc_{i}^{V})$; $Wc_{i}^{Q}$, $Wc_{i}^{K}$, $Z.Wc_{i}^{V}$ are linear embeddings for the column-attention sub-layer, and the standard self-attention operator $\text{Att}(Q, K, V) = \text{softmax}(Q.K^{T}/\sqrt{D}).V$.
> Lastly, the feed-forward sub-layer applies a row-wise feed-forward network: $\text{FF}(Z) = Z + \text{rFF}(LN(Z)) \in \mathbb{R}^{N.L.D}$.
> Finally, we make predictions for the targets of interests by feeding the corresponding target embeddings from the last layer into a
> L2-penalized linear projector, and obtain logits over the amino acid vocabulary at each position via a linear projection of the embeddings of each token in each sequence from the last layer as well.
>
> **Iterative protein redesign experiment** (Fig. B 1 in pdf)
>
> We first select an initial labeled data $D_L$, drawing points at random from the set $D$ of all mutants in the corresponding DMS assay, and keep all other points as our unlabeled pool set $D_{U} = D \backslash D_{L}$.
> At each cycle, we first train the considered model (ProteinNPT or baselines) on $D_{L}$. We then predict the property and quantify our prediction uncertainty for all possible variants in $D_{U}$. We then sequentially acquire a batch of $B$ points by greedily optimizing the Upper Confidence Bound (UCB) acquisition function $\alpha(\boldsymbol{x} ; \lambda) = \mu(\boldsymbol{x}) + \lambda \sigma(\boldsymbol{x})$, where $\lambda$ is an hyperparameter controlling the exploration/exploitation trade-off.
>
> **C3. It would be beneficial if the authors could provide a detailed description of the multi-task optimization process.**
> The same architecture and training procedure used for single property prediction directly extends to the multi-task setting by adding as many target columns as there are properties to predict (lines 178-179). There are no other modifications needed. Besides being very practical, it also allows us to capture correlation between targets as we also perform self-attention between labeled columns.
>
>
> **C4. Could you explain how the annealing process is implemented for the token prediction objective?**
> We progressively increase the relative coefficient of the target prediction objective, and reduce that of the token denoising objective. This forces the network to first learn good representations of tokens via the reconstruction objective, and then progressively focus more and more on the main task of interest.
>
> **C5. Could you elaborate on the use of a cosine schedule throughout the training process?**
> The cosine annealing  scheme allows to gradually decreasing the learning rate following a cosine curve over a predefined number of epochs, helping with convergence and has been observed to achieve strong performance in practice (see Loshchilov & Hutter, SGDR: Stochastic Gradient Descent with Warm Restarts).

---

> > ### Comment · Reviewer_8GnD · 2023-08-14
> >
> > Thanks for the clarification, I will raise the score to 5.

---

### Official Review · Reviewer_yxtV · 2023-07-08

**Soundness:** 4 excellent
**Presentation:** 3 good
**Contribution:** 2 fair
**Rating:** 5
**Confidence:** 3

**Summary:**

The authors propose to use non parametric transformers to model the ‘fitness’ landscape and the protein primary sequences with a Bert-like model. Both tokens and continuous attributes can be masked in the modeling. Additionally, the authors introduce several novel datasets to test and benchmark their model. Finally, a directed-evolution like experiment is conducted to evaluate the capability of the proposed model to be used to assist the design.

**Strengths:**

1- The authors’ problem is well motivated and clearly presented. The method is sound and particularly adapted to the tackled problem hence necessitating the conducted investigation.

2- I found the discussion and the different splits for training the model very interesting and provides additional insight on how the model behaves given novel mutations.

3- Adding novel datasets to the field in a standardized way is also in my opinion a great contribution to the bio-ML community. Nonetheless, given that there is room for additional details, I am disappointed about the lack of description of the novel datasets in the main part of the paper.

4 - The results seem rather conclusive that the method is well suited to adress the learning of the joint distribution despite several more investigations could be conducted for confirmation.

**Weaknesses:**

1 -  A notable exception in my opinion to the clarity of the paper is section 6 that i struggled to understand.

2 - There is in my opinion little discussion on how the inputs are constructed. For instance, how many proteins are included in the “alignement”. Are there any learning tradeoffs there ? For example: number of doable gradient steps before overfitting  versus  available able information in the alignment ?

3 - I found it hard to guess which baseline was what in Table 1.  Moreover, the results in table 1 does not present any variance in the experiments.

4 - Since I found the presented work interesting, I was interested in what problem could be addressed by such a model and the expected
limits (minimum number of datapoints, minimum number of datapoints in the alignments, gradient steps and so on) ?

5 - A lot of paper in the ML for protein design literature uses datasets such as GFP. Can the authors comment on their choice of not using this standard dataset.

**Questions:**

1-  How would the model behave if at inference, we specify the desired target and mask (randomly or any other fashion) some tokens ?

2 - To follow up the previous question, since your model learns a joint probability distribution, can you use this property for protein design ? I don’t believe that this is what is done in your section 6, but devising to an experiment like this would prove the ability to generate tokens conditionally to the output value.

3 - What is the influence of the number of rows in the input sequence ?

4 - Given that the current version does not exceed the 9 page limit, I strongly encourage the authors to provide either some hyper-parameterisation or some explanation on the dataset.

5 - I also found unclear in the main paper whether the authors were responsible for the experiments that led to the new datasets.

6 - How does the model scale with the number of data points ?


Note that i am willing to change my evlulation if my questions are addressed.

**Limitations:**

See Weaknesses & Questions

Minor typo: line 96

---

> ### Author Rebuttal · Authors · 2023-08-10
>
> Thank you for the very thoughtful comments and suggestions. We address each of your questions below, including several additional analyses which we believe significantly strengthen our submission.
>
> **C1. I am disappointed about the lack of description of the novel datasets in the main part of the paper.**
> Thank you for the suggestion. We provide high level information (eg, # of mutants, type of assay, taxon, MSA depth, reference sequence) for each assay in the reference file in the supplementary material (see ProteinNPT_repo/proteingym/ProteinGym2_reference_file_substitutions.csv, rows 89 -101). We will also include a new table in Appendix A.1 with a description in plain english for each assay.
>
> **C2. A notable exception in my opinion to the clarity of the paper is section 6 that I struggled to understand.**
> We have entirely rephrased the text in this section for clarity, and have included an algorithm (see Fig. B in pdf) to further clarify the approach. A detailed description of the algorithm is provided in response to reviewer 8GnD (C2).
>
> **C3. I found it hard to guess which baseline was what in Table 1. [...] Results in table 1 does not present any variance in the experiments.**
> A detailed description of baselines can be found in Appendix D1. We have added a reference to it in the caption of Table 1 to clarify, and updated Table 1 to include average standard errors across folds.
>
> **C4. Since I found the presented work interesting, I was interested in what problem could be addressed by such a model and the expected limits [...] What is the influence of the number of rows in the input sequence ?**
> We analyze your questions under two different angles, at training vs inference time. At training time, we conducted the following new analysis. We split the 100 assays in three equal-size groups depending on the number of available labels, and report the group-level performance in Table B (see pdf). We do not observe particular drops of performance in the “low depth” group compared with  groups with more labels. We interpret this phenomenon by the fact that the token denoising objective that is used during training acts as a regularization mechanism during training that confers a lot of stability to the training dynamics across a diverse set of settings, and prevents overfitting. We will update the text to clarify. For the impact of the number of  points at inference time, please refer to the ablation on this in Appendix B2 Table 6. We find that performance generally improves as we increase the number of sequences, up to a certain level (1k sequences).
>
> **C5. A lot of paper in the ML for protein design literature uses datasets such as GFP. Can the authors comment on their choice of not using this standard dataset.**
> The GFP assay is also included in our evaluation, as it was part of the original ProteinGym benchmark. In Figures 4-6 in Appendix, it corresponds to “GFP_AEQVI_Sarkisyan_2016”. These 3 figures allow us to appreciate the important variability of performance of different methods across assays. Focusing on one assay in particular, or even a handful of them, would significantly limit our ability to draw robust conclusions. To our knowledge, our work is the first one to date to benchmark protein models on that scale (at least 100 assays).
>
> **C6. How would the model behave if at inference, we specify the desired target and mask [...] I don’t believe that this is what is done in your section 6, but devising to an experiment like this would prove the ability to generate tokens conditionally to the output value.**
> Thank you for the fantastic question! We believe this is a key strength of the architecture and have run an additional analysis as suggested (Fig A of pdf). We first identify the sequence with the highest measured property for a given assay (we focused on the GFP assay in this experiment, given your comment above). We then form an input batch (randomly selecting other labeled points), mask a fixed number of tokens (5 in our experiment) in the fittest sequence, obtain the resulting log softmax over the masked positions with a forward pass and sample from these to obtain new sequences. Critically, rather than selecting the masked positions at random, we sampled them from the positions with the highest average row-attention coefficient with the target (across heads) in the last layer of ProteinNPT. This helped ensure we would select mutations at the positions with highest impact on the target of interest. We generated 1k new sequences with that process and measured the corresponding fitness with an independent zero-shot fitness predictor (ESM-1v), which we then compared with baseline in which mutations are randomly selected or selected to further minimize the fitness of the less fit protein.
>
> **C7. I also found unclear in the main paper whether the authors were responsible for the experiments that led to the new datasets.**
> No, the different assays were carried out in previously published work which we have referenced in Appendix A1. Our contribution has been to survey existing literature to identify assays that are relevant from a fitness prediction standpoint. This is a non-trivial effort that involves screening for certain standards in assay quality (eg., dynamic range, correlation between replicates), identify the experimental phenotype that is most relevant, preprocess and standardize the raw data in a way that is consistent with the other assays in the benchmark.
>
> **C8. How does the model scale with the number of data points ?**
> Since we use tied-row attention (Appendix B1) the row-attention memory footprint is simply quadratic in the sequence length but invariant to the number of rows used in each input batch. The column attention computational complexity and memory footprint is however quadratic in the number of data points and linear in sequence length.

---

> > ### Comment · Reviewer_yxtV · 2023-08-22
> > **Response to the authors**
> >
> > First of all, I thank the authors for their rebuttal that clarifies some aspects of their work. I think the generation / optimization experiment provides interesting insights on the model behave for the very difficult design task. The answer to C.4 is also particularly valuable for reproducibility and validating the approach.
> >
> > I raise my score to 5.

---

### Author Rebuttal · Authors · 2023-08-10

Dear reviewers,

We sincerely thank you for the time spent engaging with our paper and really appreciate the thoughtful comments. Based on your feedback, we have conducted additional experiments to further explore the strengths of our proposed approach, and have also clarified all points you had raised. We believe the submission is much stronger as a result. We summarize the key points of feedback and how we addressed them as follows:

1. **Novelty and contributions of this work (all reviewers)**: Based on reviews, we wanted to first restate what we believe the key contributions and novelty introduced by our work:
- **Conceptual shift over prior supervised fitness prediction models**: methods relying on embeddings extracted from large protein language models have recently led to higher performance over prior approaches (Sec. 2). However, since the token embedding dimensions are large (eg, 1280 per token for ESM-1v), these methods typically apply a pooling operator (eg, mean pooling across the full sequence length) to prevent overfitting during training, which potentially destroys valuable information for the downstream task. In contrast, ProteinNPT does not pool embeddings but leverages self-attention to learn the dependencies between labels and the embeddings of specific residues in the sequence, thereby focusing on positions that matter for the property of interest.
- **New methodological developments**: we summarize the differences between ProteinNPT and the closest model architectures in prior literature in Appendix B.6 and add further clarifications in response to reviewer 42SC (C1). We also explore, for the very first time, various aspects of non-parametric transformers such as the use of auxiliary labels (Sec. 4.4 - critical to top performance as per Table 4), the multiple property optimization setting (Sec. 5.4), conditional sampling (Sec. 4.5 and point #3 below) and uncertainty quantification (Sec. 6, Appendix G1 and point #4 below).
- **Benchmarking improvements**: we introduced 2 novel cross validation schemes to better assess the ability of fitness predictors to generalize to mutations at unseen positions (Appendix A2). We curated 13 new assays (Appendix A1) to enrich the ProteinGym benchmark collection, allowing us to compare supervised fitness prediction models on a _unprecedented_ scale in terms of number of proteins and mutants, as well as diversity of properties considered (e.g., binding affinity, enzymatic activity, thermostability)
- **Significant performance lift in all experimental settings**: our final architecture leads to _significant_ performance increase in various experimental settings: effects of single mutants, effects of multiple mutants, simultaneous prediction of multiple properties, and iterative protein redesign -- all of which are of major importance in practical protein engineering and variant effects prediction. Our models are trained on commodity hardware (a single GPU is needed) and the same hyperparameters are used across all settings and protein types, thereby facilitating its adoption by practitioners.

2. **Mathematical equations (Rev. 8GnD, af1N, 42SC)**: As a means to improve the clarity of the description of the ProteinNPT architecture, we have updated Sec. 4 to include mathematical equations that complement the main architecture diagram (Fig. 1). In particular, we hope this will clarify the _significant_ differences between our architecture and the standard “Transformer” architecture (Rev. af1N). For more details, please see the response to Rev. 8GnD (C2) where we provide the aforementioned equations.

3. **Conditional sampling for protein design (Rev. yxtV)**: We further explore the conditional sampling capabilities of ProteinNPT (Sec. 4.5) for protein engineering (Fig. A in pdf). We conduct new analyses in which we demonstrate the ability of ProteinNPT to sample new sequences with fitness significantly higher than the fittest sequence in the labeled dataset. We describe this analysis in detail in our response to Rev. yxtV (C6). This demonstrates the versatility of usages of ProteinNPT to support various protein design tasks.

4. **Uncertainty quantification (Rev. af1N, 42SC)**: Given the points raised on novelty we wanted to solidify our analysis of uncertainty quantification in non-parametric transformers and reaffirm it as a meaningful contribution of this work. We developed and compared three uncertainty quantification schemes:
- **MC dropout**: we use a fixed inference batch at inference, apply MC dropout (Gal et al., Dropout as a Bayesian Approximation) and use the standard deviation of the prediction across multiple forward passes as our uncertainty metric.
- **Batch resampling**: for a given point we want to quantify the uncertainty of, we perform several forward passes by completing the input batches with a different sample (with replacement) of training points for each pass. No dropout is applied and we use the standard deviation across forward passes as our uncertainty metric.
- **Combined scheme**: we combine the first two schemes together (ie, turning on dropout in the batch resampling sampling)
The last scheme delivers superior performance in calibration curve analyses (Fig. C), and we therefore refreshed the results of our iterative design analyses to use ProteinNPT in conjunction with this uncertainty quantification scheme instead, subsequently leading to even higher performance on this task (Fig. D).

In addition to this overall response, we provide detailed responses to all comments raised by each reviewer. Please reach out to us if you would like us to clarify any remaining points.

---

### Decision · Program_Chairs · 2023-09-21

**Decision:**

Accept (poster)

**Comment:**

The paper proposes a transformer model trained with masked-language modeling for protein prediction and design tasks. The evaluation is quite extensive and the proposed ProteinNPT performs very well as shown.
All reviewers liked the paper and the authors did a great job in addressing all the reviewer concerns and questions.